# Neural Operators for Multi-Task Control and Adaptation

## Abstract

Neural operator methods have emerged as powerful tools for learning mappings between infinite-dimensional function spaces, yet their potential in optimal control remains largely unexplored. We focus on multi-task control problems, whose solution is a mapping from task description (e.g., cost or dynamics functions) to optimal control law (e.g., feedback policy). We approximate these solution operators using a permutation-invariant neural operator architecture. Across a range of parametric optimal control environments and a locomotion benchmark, a single operator trained via behavioral cloning accurately approximates the solution operator and generalizes to unseen tasks, out-of-distribution settings, and varying amounts of task observations. We further show that the branch–trunk structure of our neural operator architecture enables efficient and flexible adaptation to new tasks. We develop structured adaptation strategies ranging from lightweight updates to full-network fine-tuning, achieving strong performance across different data and compute settings. Finally, we introduce meta-trained operator variants that optimize the initialization for few-shot adaptation. These methods enable rapid task adaptation with limited data and consistently outperform a popular meta-learning baseline. Together, our results demonstrate that neural operators provide a unified and efficient framework for multi-task control and adaptation.

## 1 Introduction

In many control applications, one must solve not a single optimization problem but a family of related ones, such as navigating to different goal locations, tracking reference trajectories under varying vehicle parameters, or planning paths through environments with different obstacle and terrain configurations. Each such variation defines a distinct task, specified by the choice of dynamics, cost function, or constraints that characterize the control problem. The fundamental challenge is that even modest changes in these specifications can induce substantially different optimal policies, so a successful multi-task approach must capture how changes in task specifications translate into changes in the desired policy. This relationship is often highly nonlinear, making naive parameter-sharing or task-conditioning strategies insufficient in many practical scenarios. Furthermore, when the amount of available task data varies, some tasks may have many expert demonstrations while others have few. The learned model must be robust to this heterogeneity. This structure suggests that multi-task control is naturally formulated as a mapping between function spaces: from task-defining functions to optimal policies. This perspective motivates our use of neural operators, which are designed to approximate mappings between function spaces and are thus a natural model class for multi-task control.

From this perspective, common approaches to multi-task control can be viewed as approximations to the underlying structure of the problem: a mapping between infinite-dimensional function spaces. Methods that learn a separate policy per task (Teh et al., 2017; Haldar & Pinto, 2023) ignore the shared structure across tasks and scale poorly. Task-embedding approaches represent tasks as finite-dimensional vectors, but their performance can be sensitive to architectural choices and variability in the number of data points for the task-defining function. Meta-learning methods such as Model-Agnostic Meta-Learning MAML (Finn et al., 2017) learn a parameter initialization that enables rapid adaptation to new tasks with few gradient steps. Unlike neural operators, however, MAML does not explicitly model the mapping between function spaces. In contrast, neural operators are designed to learn mappings between function spaces directly, enabling zero-

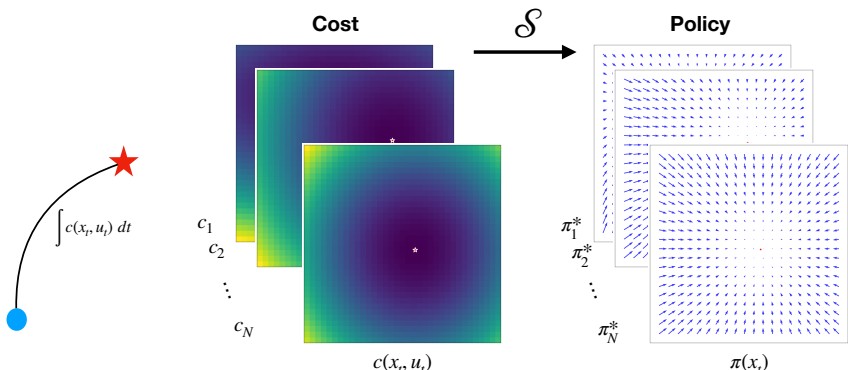

Figure 1: A point-to-point multi-task control problem. **Left:** A single point-to-point task. **Middle:** A series of tasks represented by their cost functions $c_i$. **Right:** The corresponding optimal policies $\pi_i^*$ where the small arrows correspond to the control outputs at different x,y positions. This is naturally modeled as a mapping between function spaces.

shot predictions at inference time. This makes them particularly well suited for settings where new policies must be produced quickly across many tasks at deployment.

Neural operators have been successfully applied across a wide range of scientific computing problems, including weather forecasting, fluid dynamics, and materials science (Kovachki et al., 2023; Nghiem et al., 2023; Lu et al., 2021; Pathak et al., 2022; Tretiakov et al., 2025; Li et al., 2020a), supported by universal approximation guarantees for continuous operators between Banach spaces (Chen & Chen, 1995). A useful property of several neural operator (NO) architectures is their ability to represent complex, nonlinear mappings between function spaces while remaining invariant to both the cardinality and ordering of samples in the input. Despite this natural fit, their application to control problems remains largely unexplored. In this work, we investigate the SetONet architecture (Tretiakov et al., 2025), built on DeepONet (Lu et al., 2021), as a model for multi-task control. We train an operator via behavioral cloning and show that it accurately approximates the mapping from task-defining functions to optimal policies.

In practice, the pretrained operator may encounter tasks that differ from those seen during training. To handle this, we develop several adaptation strategies that exploit the branch–trunk decomposition of SetONet, ranging from last-layer updates to full-network fine-tuning. We further propose two meta-training variants, SetONet-Meta and SetONet-Meta-Full, which optimize the operator initialization for rapid few-shot adaptation via a bi-level objective inspired by MAML (Finn et al., 2017). We evaluate all methods on four parametric optimal control environments and a locomotion task from the iMuJoCo benchmark (HalfCheetah-v3) (Patacchiola et al., 2023). Our main contributions are:

- **Neural operators for multi-task control.** We establish neural operators as an effective model for multi-task control, a setting that has been relatively underexplored in the neural operator literature. We demonstrate through a series of experiments on environments of varying complexity that NOs can accurately approximate the solution operator over a distribution of tasks. Moreover, because the architecture operates on variable-size, unordered task data, the learned operator generalizes across amounts of data not seen during training.

- **Structured adaptation for new tasks.** While the pretrained operator performs well on tasks near the training distribution, accuracy degrades on out-of-distribution tasks or when training data is limited. We show that the neural operator architecture enables a spectrum of efficient adaptation strategies, from lightweight last-layer updates to full-network fine-tuning. Partial fine-tuning achieves accuracy comparable to full-network fine-tuning at a fraction of the cost, and cost-based fine-tuning allows adaptation without any explicit expert demonstrations.

- **Meta-trained operators for few-shot adaptation.** We propose two novel meta-training variants that optimize the operator initialization for rapid adaptation. SetONet-Meta restricts the inner loop to a subset of model parameters, providing data-efficient updates that are especially effective when pretraining data is limited or adapting to tasks that are close to the training distribution. SetONet-Meta-Full is a model that adapts the full set of model parameters, enabling rapid generalization to tasks outside the training distribution. Each of the meta-training models has trade-offs depending on initial training data, online compute limitations, and whether the downstream task is OOD. Both variants consistently outperform the MAML baseline across all environments.

## 2 Related Work

### 2.1 Data-Driven Optimal Control

Optimal control plays a foundational role in modeling and decision-making for complex dynamical systems, with widespread applications in robotics, aerospace, and process engineering (Betts, 2010; Bertsekas, 2012; Rawlings et al., 2020). Classical approaches such as LQR and MPC provide strong theoretical guarantees, but rely on repeatedly solving computationally intensive optimization problems online, which can limit their scalability to high-dimensional and highly nonlinear systems. These challenges have motivated learning-based and data-driven approaches that aim to approximate optimal control policies through offline learning, effectively amortizing the cost of repeated online optimization. A prominent strategy is to leverage the demonstrations generated by expert controllers and to recast control as a supervised learning problem (Hertneck et al., 2018; Karg & Lucia, 2020; Chen et al., 2018). This paradigm enables faster inference at runtime than classical methods and is often more suitable for real-time control tasks. Leveraging known dynamics and expert demonstrations, training the policy model can often be done more efficiently and accurately compared to model-free reinforcement learning (Reddy et al., 2019).

While offline imitation amortizes the cost of online optimization, the resulting policy is fundamentally limited by the expert demonstrations it is trained to reproduce. A complementary line of work therefore refines an offline-trained policy against the task objective itself, paralleling the offline-to-online paradigm in reinforcement learning, where an offline-initialized policy is improved through subsequent online interaction (Ball et al., 2023). When the dynamics are differentiable, this refinement can be carried out without collecting new data: the control objective is backpropagated through closed-loop rollouts to obtain analytic policy gradients, as in differentiable predictive control (Drgoňa et al., 2024). Our cost-based fine-tuning (Section 4.2) instantiates this idea for neural operators: starting from a behaviorally-cloned operator, it minimizes the control objective along differentiable rollouts, requiring no supervisory controller and adapting a pretrained operator rather than training a policy from scratch. This also connects to recent self-supervised operator learning for control (Xu et al., 2025).

### 2.2 Multi-Task Control

Learning a controller that can adapt to different tasks remains a challenge. When the total number of tasks is limited, a common approach, particularly in reinforcement learning, is to learn a separate policy for each task. This is often followed by a *distillation* step, in which information shared across tasks is consolidated into a single policy, as proposed in (Teh et al., 2017). Variants of this distillation paradigm have been shown to improve performance by explicitly conditioning the final policy model with its task label (Haldar & Pinto, 2023). Although effective when the number of tasks is small and sufficient training data is available, such approaches struggle to scale: the computational cost of training and storing separate policies becomes prohibitively high for complex problems. Neural operators sidestep these scalability issues by learning a single mapping from task descriptions to policies, avoiding the need to train or store separate models as the number of tasks grows.

Given such limitations, it is often preferable to learn a central policy that can perform well across a family of tasks (Ammar et al., 2014; Deisenroth et al., 2014). For parametric optimal control problems in which each task is uniquely defined by a set of problem-specific parameters, Drgoňa et al. (2024) propose augmenting the policy's state inputs with vectors of explicit task parameterizations. Although straightforward to imple-

ment, this assumption can be difficult to satisfy in practice, particularly in reinforcement learning settings where accurate task models are not always accessible. A common alternative is to introduce learned task representations (Sodhani et al., 2021; Humplik et al., 2019; Rakelly et al., 2019; Marza et al., 2024; Lan et al., 2019; Hansen et al., 2023). These approaches typically learn a task embedding separately from the policy, mapping task-relevant information to a fixed-dimensional vector. These learned representations can then be used in the policy training as conditioning explicitly or implicitly. Although effective in many settings, this formulation introduces a representational bottleneck: task descriptions must be compressed into a fixed-length vector, making performance sensitive to observation resolution, ordering, and task coverage. How task information is utilized, is itself a consequential design choice: naive strategies such as concatenating the task representation to the network input are often insufficient, which has motivated a range of alternatives, from feature-wise modulation (Perez et al., 2018) to attention- and operator-based conditioning (Wang et al., 2024b). This is precisely the axis along which the set-encoder branch of SetONet differs from a fixed-length concatenation: it consumes a variable-size, permutation-invariant context rather than a flattened vector, avoiding the resolution- and ordering-sensitivity of naive conditioning. Moreover, the assumptions required for convergence guarantees and other theoretical results are often quite restrictive (Tutunov et al., 2018).

More recently, set-based and attention-based architectures have been explored to handle variable-size task inputs in reinforcement learning. Mern et al. (2020) propose an attention-based input representation that is invariant to the ordering and number of objects in the observation, improving sample efficiency in environments with exchangeable entities. Zhou et al. (2022) formalize this setting through entity-factored Markov decision processes (MDPs) and show that Deep Set and Self-Attention policy architectures enable compositional generalization to varying numbers of entities at test time. These approaches share some properties with the SetONet model used in our work; namely, a permutation-invariant encoder that accommodates variable-size inputs. However, they operate at the level of state representations and policy conditioning rather than learning a mapping between function spaces. The operator methods used in our work provide a principled framework for learning the infinite-dimensional mappings between tasks and control policies.

Lastly, meta-learning provides an alternative paradigm for multi-task control, achieving task adaptation not through architecture design but through a training objective that explicitly optimizes for rapid adaptation from few examples. Methods such as MAML (Finn et al., 2017) and its variants (Collins et al., 2020; Barman et al., 2024) aim to learn a policy initialization from which task-specific policies can be obtained with limited data and a small number of gradient steps, without requiring explicit task representations. While conceptually appealing, these approaches incur additional computation overhead at online deployment, limiting their use in real-time control settings, where fast and reliable adaptation is critical. In contrast to these approaches, neural operators offer a principled approach that avoids both the representational bottleneck of fixed-dimensional task embeddings and the computational overhead of gradient-based adaptation at deployment. In Section 5.4, we empirically demonstrate these advantages by comparing NO based policies against a MAML baseline across several adaptation scenarios, including zero-shot generalization and rapid online fine-tuning.

## 2.3 Neural Operators

Neural operators (NOs) have emerged as a principled approach for learning mappings between infinite-dimensional function spaces, offering strong theoretical support (Kovachki et al., 2023; 2021; Lu et al., 2021) and robust empirical performance across a wide range of differential equations and broader scientific computing problems (Azizzadenesheli et al., 2024; Choi et al., 2024; Rashid et al., 2022; Li et al., 2025). By formulating task adaptation directly as an operator learning problem, neural operators naturally align with multi-task optimal control settings and enable efficient task-dependent adaptation with minimal online computational overhead. Despite this potential, their application to control problems remains relatively underexplored. Recent work has shown that solution operators can be learned in an unsupervised manner for mean-field games (Huang & Lai, 2024), with further error analysis and demonstrations of viability for certain open-loop control problems provided in Xu et al. (2025). However, much of the existing literature still focuses on single-task control formulations (Bhan et al., 2023), highlighting the need for broader investigation of neural operators as policy learners for general parametric control problems. Additionally, while comparisons and integrations of MAML-style meta-learning and neural operators have been explored in simulation and

design contexts (Wang et al., 2024a), their relative advantages and trade-offs in control applications remain poorly understood. Our work aims to address these gaps.

## 3 Preliminaries

In this section, we formulate the class of parametric optimal control problems that defines the multi-task control setting (Section 3.1), discuss imitation learning: a method for approximating optimal policies given expert demonstrations (Section 3.2) and introduce the neural operator architectures DeepONet and SetONet (Section 3.3).

### 3.1 Parametric Optimal Control Problems

Consider a dynamical system with state $\mathbf{x}_t \in X \subset \mathbb{R}^{d_\mathbf{x}}$ and control input $\mathbf{u}_t \in U \subset \mathbb{R}^{d_\mathbf{u}}$ at discrete times $t \in [T] := \{0, 1, \ldots, T\}$. Let $\Phi \subset \mathbb{R}^{d_\phi}$ and $\Psi \subset \mathbb{R}^{d_\psi}$ denote the sets of cost and dynamics parameters. Each task is specified by a pair $(\phi, \psi) \in \Phi \times \Psi$, where $\phi$ determines the stage and terminal costs $(\ell, \ell_T)$ and $\psi$ determines the dynamics $\mathbf{g}$. We seek a feedback policy $\pi\colon X \times [T-1] \times \Phi \times \Psi \to U$ that solves the following discrete-time parametric optimal control problem (pOCP):

$$\pi^*(\,\cdot\,; \phi, \psi) \in \arg\inf_\pi \ \mathbb{E}_{\mathbf{x}_0 \sim P_{\mathbf{x}_0}} \left[ \sum_{t=0}^{T-1} \ell\big(\mathbf{x}_t, \pi(\mathbf{x}_t, t; \phi, \psi), t; \phi\big) + \ell_T(\mathbf{x}_T; \phi) \right], \tag{1}$$

where $P_{\mathbf{x}_0}$ denotes some known probability distribution over the initial state $\mathbf{x}_0$, the state trajectory evolves according to the closed-loop dynamics

$$\mathbf{x}_{t+1} = \mathbf{g}\big(\mathbf{x}_t, \pi(\mathbf{x}_t, t; \phi, \psi), \psi\big) \quad \forall\, t \in [T-1], \tag{2}$$

and the control input at time $t$ is generated by the feedback policy $\pi$, i.e.

$$\mathbf{u}_t = \pi(\mathbf{x}_t, t; \phi, \psi) \quad \forall\, t \in [T-1]. \tag{3}$$

For any fixed task $(\phi, \psi)$, and initial fixed state $\mathbf{x}_0$, the objective (1) together with (2) and (3) define a standard finite-horizon optimal control problem, which can be solved via classical approaches, e.g. shooting or collocation (Jacobson & Mayne, 1970; Hargraves & Paris, 1987). Although each task is generated by a finite-dimensional parameter, we assume that there is no direct access to $(\phi, \psi)$ and instead observe only pointwise evaluations of the induced cost or dynamics functions. Across the family of tasks, the resulting solutions define a mapping $\mathcal{S}$ from the task-defining functions parameterized by $(\phi, \psi)$ to the corresponding optimal feedback law $\pi^* \in \Pi = \{\pi^*(\,\cdot\,; \phi, \psi) : (\phi, \psi) \in \Phi \times \Psi\}$; this is precisely the object we will aim to approximate with neural operators. To isolate the effects of cost and dynamics on the solution operator, we vary only one set of parameters at a time, with $\phi \sim P_\Phi$ and $\psi \sim P_\Psi$ sampled independently. We therefore focus on approximating the following two solution operators:

$$\mathcal{S}_\psi : \mathcal{F} \to \Pi, \qquad \mathcal{S}_\phi : \mathcal{G} \to \Pi, \tag{4}$$

For $\mathcal{S}_\psi$, we assume fixed dynamics parameters $\psi$, and for $\mathcal{S}_\phi$, fixed cost parameters $\phi$. The cost function space is denoted $\mathcal{F} = \{(\ell(\,\cdot\,; \phi), \ell_T(\,\cdot\,; \phi)) : \phi \in \Phi\}$ and the dynamics function space $\mathcal{G} = \{\mathbf{g}(\,\cdot\,; \psi) : \psi \in \Psi\}$. Note that even though the input function space for $\mathcal{S}_\psi$ does not depend on the dynamics of the system, $\Pi$ always does. Here, we assume that $\mathcal{F}$ and $\mathcal{G}$ are Banach spaces. Concretely, $\mathcal{S}_\psi$ maps from $\mathcal{F}$ onto $\{\pi^*(\,\cdot\,; \phi, \psi) : \phi \in \Phi\} \subset \Pi$, and $\mathcal{S}_\phi$ maps from $\mathcal{G}$ onto $\{\pi^*(\,\cdot\,; \phi, \psi) : \psi \in \Psi\} \subset \Pi$. The setting in which the input and output function spaces are accessible only through finite collections of pointwise samples is precisely the regime addressed by neural operator architectures, which we introduce in Section 3.3.

### 3.2 Imitation Learning

Given access to task functions $\ell(\,\cdot\,; \phi), \ell_T(\,\cdot\,; \phi)$ and $\mathbf{g}(\,\cdot\,; \psi)$ and an expert solver for the parametric optimal control problem (1), we can generate demonstrations consisting of $m$ state–action pairs

$\{(\mathbf{x}_j, t_j, \pi^*(\mathbf{x}_j, t_j; \boldsymbol{\phi}, \boldsymbol{\psi}))\}_{j=1}^m$ where $\mathbf{u}_j = \pi^*(\mathbf{x}_j, t_j; \boldsymbol{\phi}, \boldsymbol{\psi})$ for a particular task $(\boldsymbol{\phi}, \boldsymbol{\psi})$. Here we use $\pi^*$ to denote the optimal policy to the problem. *Behavioral cloning* (BC) (Pomerleau, 1988), is a widespread technique for training a parametric policy $\hat{\pi}_\theta$, by minimizing the following loss:

$$\mathcal{L}_{\mathrm{BC}}(\theta) = \frac{1}{m} \sum_{j=1}^m \left\| \hat{\pi}_\theta(\mathbf{x}_j, t_j, \boldsymbol{\phi}, \boldsymbol{\psi}) - \pi^*(\mathbf{x}_j, t_j; \boldsymbol{\phi}, \boldsymbol{\psi}) \right\|^2. \tag{5}$$

In the single-task setting, this results in a policy that mimics the expert for one fixed choice of $(\boldsymbol{\phi}, \boldsymbol{\psi})$. This form of imitation learning is appealing because it reduces the problem of learning a policy to a supervised learning problem.

In the *multitask* setting, demonstrations are collected across a family of tasks; for example, $\boldsymbol{\phi}_i \sim P_\Phi$ for $i = 1, \ldots, N$ with $\boldsymbol{\psi}$ held fixed. The goal is then to learn a single model that, given the information identifying the current task, produces a task-specific policy. A naïve approach is to train a separate policy per task, but this scales poorly with the number of tasks and does not exploit any shared structure in the family of tasks. Alternatively, a single policy can be conditional on a task representation, but this introduces challenges discussed in Section 2.2. The operator learning perspective developed in Section 4.1 offers a principled alternative: rather than conditioning on a finite-dimensional task embedding, the model receives pointwise evaluations of the task-defining function and learns a mapping to a space of expert policies. This multi-task behavior cloning approach retains the simplicity of (5) while still being able to learn the shared structure across tasks.

### 3.3 Neural Operators

In this work, we will approximate the solution operators of (4) primarily using SetONet (Tretiakov et al., 2025), a neural operator architecture that builds on DeepONet (Lu et al., 2021). DeepONet is grounded in the universal approximation theorem for operators (Chen & Chen, 1995), which establishes that continuous operators between Banach spaces can be approximated to arbitrary accuracy by a neural network comprising two sub-networks called the *branch network* and *trunk network*. SetONet inherits this theoretical foundation and additionally introduces a permutation-invariant set encoder in the branch network, making it naturally compatible with variable-sized, unordered input data, properties that are particularly advantageous in control settings where the number and arrangement of task observations may vary. Other popular operator learning approaches such as Fourier Neural Operators (FNO) (Li et al., 2020a) and Graph Kernel Networks (Li et al., 2020b) impose spectral or locality-based inductive biases that, while effective for many PDE problems, are less clearly motivated in parametric optimal control.

Here we briefly describe the SetONet setup and architecture. Consider the solution operator $\mathcal{S}_{\boldsymbol{\psi}} : \mathcal{F} \rightarrow \Pi$ (the dynamics-varying case is analogous). An input function $\ell_i \in \mathcal{F}$ is observed only by pointwise evaluations at a finite number of *context points* $\mathbf{x} \in \mathbb{R}^{d_x}$, and the output expert policy $\pi^* \in \Pi$ is predicted at a finite number of *query locations* $\mathbf{y} \in \mathbb{R}^{d_y}$. The input and output functions may require multiple inputs as context $(\mathbf{x}_t, \mathbf{u}_t, t)$, which we omit for brevity in this section. SetONet comprises a *trunk network*, which learns a collection of $p$ basis functions $\{b_1, b_2, \ldots, b_p\}$ in the policy space $\Pi$, and a *branch network* that maps a variable-sized set of input function evaluations to the associated coefficients $\{c_1, c_2, \ldots, c_p\}$ via a permutation-invariant set encoder. The basis functions $b_k$ may be vector-valued (e.g., $b_k \in \mathbb{R}^{d_u}$ for control outputs) and the coefficients $c_k \in \mathbb{R}$. The operator is approximated as:

$$\mathcal{S}_{\boldsymbol{\psi}}(\ell_i)(\mathbf{y}) \approx \mathcal{T}_\theta(\ell_i)(\mathbf{y}) = \sum_{k=1}^p c_k(\ell_i) \, b_k(\mathbf{y}), \tag{6}$$

where $\theta$ are the learned parameters of both the branch network ($\theta_{branch}$) and the trunk network ($\theta_{trunk}$). Labels come in the form of samples from the input and output function spaces $\ell_i \in \mathcal{F}$ and $\pi_i^* \in \Pi$ evaluated at a finite number of sensor locations and query locations. Concretely, for each training instance $i$ we assume access to (i) a *context set* of samples of an input function $\ell_i \in \mathcal{F}$ at $m_i$ different sensor locations $\{\mathbf{x}_{ij}\}_{j=1}^{m_i}$,

$$C_i = \{(\mathbf{x}_{ij}, \ell_i(\mathbf{x}_{ij}))\}_{j=1}^{m_i}, \tag{7}$$

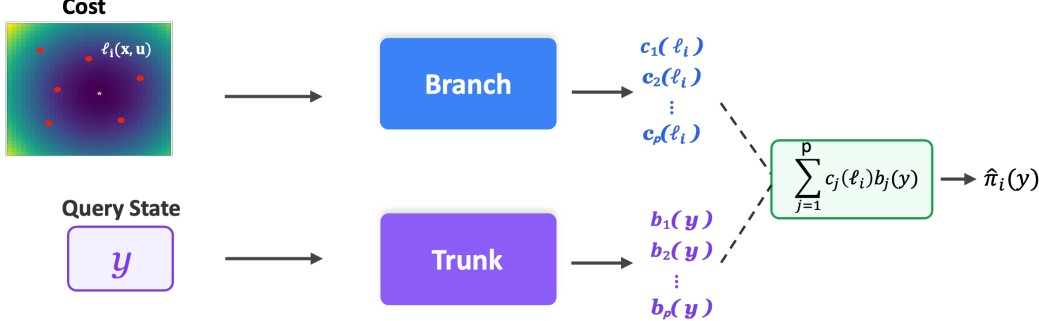

Figure 2: **DeepONet/SetONet architecture:** Here we show the mapping $\mathcal{T}_\theta[\ell_i] \to \hat{\pi}_i$, with pointwise evaluations of $\ell_i$ (in red) and of $\hat{\pi}_i$ at the point $\mathbf{y}$. The branch network maps sensor locations $(x, u)$ of a cost function $\ell(x, u; \boldsymbol{\phi})$ to task-dependent coefficients $\{c_k(\ell)\}_{k=1}^p$. The red points indicate the pointwise samples of $\ell_i$. The trunk maps query locations $\mathbf{y} = (x, t)$ to a set of learned basis functions $\{b_k(\mathbf{y})\}_{k=1}^p$. Their inner product yields the predicted control output at the query location $\mathbf{y}$

and (ii) targets consisting of evaluations of the output policy $\pi_i^* = \mathcal{S}_{\boldsymbol{\psi}}(\ell_i) \in \Pi$ at $n_i$ different query locations $\{\mathbf{y}_{ik}\}_{k=1}^{n_i}$, e.g.,

$$\{(\mathbf{y}_{ik}, \pi_i^*(\mathbf{y}_{ik}))\}_{k=1}^{n_i}. \tag{8}$$

The neural operator $\mathcal{T}_\theta$ is trained by minimizing the following empirical MSE over $K$ pairs of input-output functions $\{(\ell_i, \pi_i^*)\}_{i=1}^K$:

$$\mathcal{L}(\theta) = \frac{1}{K} \sum_{i=1}^K \frac{1}{n_i} \sum_{k=1}^{n_i} \|\mathcal{T}_\theta(\ell_i)(\mathbf{y}_{ik}) - \pi_i^*(\mathbf{y}_{ik})\|_2^2 \tag{9}$$

This decomposition is illustrated in Figure 2 for the case where $\mathcal{F}$ is the space of cost functions and the output space $\Pi$ is the space of optimal control policies. Because the branch network uses a set encoder, predictions are invariant to the ordering of samples in $C_i$ and the context set size $m_i$ may vary freely across tasks and over time. There are no restrictions on the query locations, which can be arbitrarily chosen at training or test time. Although the expert trajectories used in our experiments are generated by open-loop solvers, the operator learns a feedback mapping conditioned on the current state. Along the expert trajectory, the open-loop and closed-loop representations coincide, and the additional state dependence allows the learned policy to generalize beyond the nominal trajectory.

## 4 Methodology

We now describe how the behavioral cloning objective of Section 3.2 and the neural operator approach of Section 3.3 combine to produce a practical method for multi-task control. In Section 4.1, we show how the SetONet architecture can be trained to approximate the solution operators defined in Section 3.1 using expert demonstrations collected across a distribution of tasks. In principle, the pretrained operator can predict policies for new tasks in a single forward pass. In practice, however, approximation error, limited training data, and distributional shift between training and deployment tasks can degrade zero-shot predictions. We therefore develop strategies for adapting the pretrained operator to new tasks at deployment: fine-tuning with expert demonstrations, fine-tuning with cost feedback when expert demonstrations are unavailable (Section 4.2), and a meta-training procedure that explicitly optimizes the operator for rapid few-shot adaptation (Section 4.3).

### 4.1 Multi-Task Behavioral Cloning via Operator Learning

We now instantiate the behavioral cloning objective of Section 3.2 within the operator learning approach of Section 3.3 to approximate the solution operators $\mathcal{S}_{\boldsymbol{\psi}}$ and $\mathcal{S}_{\boldsymbol{\phi}}$ defined in (4).

**Data generation.** For concreteness, we will consider the cost-varying setting in which the dynamics $\psi$ are fixed and $\phi_i \sim P_\Phi$ for $i = 1, \dots, N$ (the dynamics-varying case is analogous). For each sampled task $\phi_i$, we obtain an expert policy $\pi_i^{\text{exp}} \approx \pi_i^*$ using a suitable optimal control solver (see Section 5.1 for specific solvers used) or RL algorithm (Soft Actor Critic (SAC) for iMuJoCo). We write $\pi_i^{exp}$ to make clear that the policy we are sampling may not exactly match $\pi_i^*$. In the OCP environments, $\pi_i^{\text{exp}}$ closely approximates the optimal feedback law; in the iMuJoCo environments, it is a learned policy that may be suboptimal. We consider each of these solvers as producing our expert training target $\pi_i^{\text{exp}}$. For brevity, we write $\ell_i$ for the cost function of the $i$-th task, encompassing both stage $\ell_i(\mathbf{x}_t, \mathbf{u}_t, t)$ and terminal $\ell_i(\mathbf{x}_t, T)$ costs $\big(\ell(\,\cdot\,;\phi_i), \ell_T(\,\cdot\,;\phi_i)\big)$. Each training instance is thus a pair of functions: a task-defining input $\ell_i$ and the corresponding expert policy $\pi_i^{\text{exp}}$.

**Neural operators for behavioral cloning.** Instantiating the SetONet model of Section 3.3, we construct for each task $i$:

1. A *context set* $C_i = \{((\mathbf{x}_{ij}, \mathbf{u}_{ij}, t_{ij}), \ell_i(\mathbf{x}_{ij}, \mathbf{u}_{ij}, t_{ij}))\}_{j=1}^{m_i}$ of pointwise cost evaluations at $m_i$ sensor locations, which the branch network uses to encode the task identity without requiring access to $\phi_i$.

2. Supervised targets $\{(\mathbf{y}_{ik}, \pi_i^{\text{exp}}(\mathbf{y}_{ik}))\}_{k=1}^{n_i}$ of expert policy evaluations at $n_i$ query locations, where $\mathbf{y}_{ik} = (x_{ik}, t_{ik})$.

The context set structure above applies directly to the cost-varying setting, but our model accommodates other task specifications with only a change in what the context encodes. In the dynamics-varying setting (with cost $\phi$ fixed, $\psi_i \sim P_\Psi$), the context set consists of dynamics evaluations $C_i = \{((\mathbf{x}_{ij}, \mathbf{u}_{ij}), \mathbf{g}(\mathbf{x}_{ij}, \mathbf{u}_{ij}; \psi_i))\}_{j=1}^{m_i}$, where each context point is a state–control pair and the observed value is the resulting next state. While we focus on specific representations of the input function for the cost-varying and dynamics-varying settings described above, we note that the choice of context encoding is not unique. Alternative representations, whether explicit or implicit, can be used provided they sufficiently characterize the task. Similarly, for a reference tracking scenario, an implicit representation of cost may be given by a set of waypoint locations. In all cases, the supervised targets for the output function space remain expert policy evaluations, the branch network receives the task-defining context, and the trunk network receives states and times at which the policy is to be predicted. The complete details for each environment are given in Section 5.2.

We approximate the operator $\mathcal{S}_\psi \approx \mathcal{T}_\theta$ by minimizing the empirical loss (9) on $N$ sets of task contexts. Then, given a new context set $C_{i'}$, $\mathcal{T}_\theta$ predicts the corresponding feedback policy at arbitrary query locations: $\mathcal{T}_\theta(\ell_{i'})(\mathbf{y}) \approx \pi_{i'}^{\text{exp}}(\mathbf{y})$ or $\mathcal{T}_\theta(\mathbf{g}_{i'})(\mathbf{y}) \approx \pi_{i'}^{\text{exp}}(\mathbf{y})$ when the input function is the space of dynamics functions. At test time, this amounts to approximating an expert policy on an unseen task via a single forward pass, without re-solving the underlying optimal control problem (1).

The formulation above and in Section 3.1 assumes that the context set consists of pointwise evaluations of a known input function. In practice, this assumption can be relaxed. For example, in our obstacle avoidance experiments the context set encodes obstacle geometry rather than evaluations of a smooth cost function, and in the iMuJoCo environments the context set consists of state-action transition samples rather than evaluations of a known dynamics model. In both cases, the branch network receives a set of tuples that implicitly characterize the task, and the architecture operates identically.

## 4.2 Task-Specific Adaptation

In the multi-task setting of Section 4.1, the operator $\mathcal{T}_\theta$ is trained on tasks sampled from distributions $P_\Phi$ and $P_\Psi$. At deployment, we may encounter a task that was not seen during training but is drawn from the same distribution. The operator $\mathcal{T}_\theta$ trained in Section 4.1 provides a global approximation to the solution operator across the task distribution. When training data is limited or the target task differs substantially from the training distribution, the pretrained operator alone may not provide sufficient accuracy. In such cases, we can adapt the operator to the new task using a small amount of task-specific data. We consider two general adaptation settings depending on what information is available for the new task.

**Adaptation with expert demonstrations.** When a small number of expert state–action pairs $\mathcal{D}_i = \{(\mathbf{y}_j, \mathbf{u}_j^*)\}_{j=1}^{n_i}$ are available for the target task, we can refine the operator by minimizing the imitation loss over these demonstrations. Given a new task $i$ with sensor locations $\mathcal{C}_i$, we minimize

$$\theta^* = \arg\min_\theta \frac{1}{n_i} \sum_{(\mathbf{y}_j, \mathbf{u}_j^*) \in \mathcal{D}_i} \left\| \mathcal{T}_\theta(f_i)(\mathbf{y}_j) - \mathbf{u}_j^* \right\|^2, \tag{10}$$

where $f_i$ is encoded via $\mathcal{C}_i$. This reduces adaptation to a supervised fine-tuning problem: the pretrained operator provides a warm start, and the demonstrations steer it toward the target task's policy.

**Adaptation with cost feedback.** In settings where expert demonstrations are unavailable but the task-specific cost function and dynamics model are known, we can bypass the behavioral cloning objective entirely and instead fine-tune the operator by directly minimizing the control objective over unrolled trajectories. Given differentiable dynamics $\mathbf{x}_{t+1} = \mathbf{g}(\mathbf{x}_t, \mathbf{u}_t; \boldsymbol{\psi})$ and cost $\ell(\mathbf{x}_t, \mathbf{u}_t, t; \boldsymbol{\phi})$, we roll out the current policy $\hat{\pi}_\theta = \mathcal{T}_\theta(\ell)$ from a batch of initial states $\{\mathbf{x}_0^{(k)}\}$ and minimize the total trajectory cost:

$$\mathcal{L}_{\text{cost}}(\theta) = \frac{1}{M} \sum_{m=1}^M \left[ \sum_{t=0}^{T-1} \ell\big(\mathbf{x}_t^{(m)}, \hat{\pi}_\theta(\mathbf{x}_t^{(m)}, t), t; \boldsymbol{\phi}\big) + \ell_T\big(\mathbf{x}_t^{(m)}; \boldsymbol{\phi}\big) \right], \tag{11}$$

where each trajectory is obtained by rolling out the policy from the initial state $\mathbf{x}_0^{(m)}$ through the differentiable dynamics, and $M$ is the number of initial conditions sampled. Gradients of (11) with respect to $\theta$ are computed via backpropagation through the entire rollout. This yields an analytic policy gradient: the control objective is differentiated through the dynamics themselves. In this sense, cost-based fine-tuning is not a separate technique but the same adaptation framework with the imitation loss (10) replaced by the control objective (11); this situates it within the offline-to-online line of data-driven optimal control discussed in Section 2.1. This is particularly appealing for tasks where the cost structure is known but expert solutions are unavailable or expensive to obtain, as it leverages the pretrained policy as a warm start and refines it using only cost feedback and a dynamics model, bypassing the difficulty of training the policy from scratch.

**Which parameters to adapt.** Recall that $\mathcal{T}_\theta(\ell_i)(\mathbf{y}) = \sum_{k=1}^p c_k(\ell_i) \, b_k(\mathbf{y})$, where the trunk produces basis functions $\{b_k\}$ shared across all tasks and the branch produces task-dependent coefficients $\{c_k\}$. The branch–trunk decomposition admits a spectrum of fine-tuning strategies that range from full parameter updates to partial retraining of the policy model (Zhu et al., 2023; Goswami et al., 2022; Xu et al., 2023; Wu, 2024; Zhang et al., 2024). We consider three strategies, listed in decreasing order of adaptation capacity:

- **Full-network fine-tuning.** As a baseline, we update all trainable parameters $\theta = (\theta_{\text{branch}}, \theta_{\text{trunk}})$ during adaptation. Full updates typically yield the lowest error but incur the highest computational cost and are most susceptible to overfitting when few demonstrations are available. In subsequent sections we refer to this as **SetONet-FT**.

- **Branch-only fine-tuning.** If the target policy lies approximately in the span of the learned basis functions, $\pi(\cdot) \approx \text{span}\{b_1, \ldots, b_p\}$, it suffices to retrain only the branch network while keeping the trunk fixed. This restricts adaptation to finding new coefficients that best represent the target task's policy in the existing basis, reducing both the number of trainable parameters and the risk of catastrophic forgetting. This strategy follows naturally from the operator fine-tuning literature (Goswami et al., 2022; Zhang et al., 2024) and mirrors the common practice in robotics of fine-tuning a task-specific head on top of a frozen backbone (Brohan et al., 2024; Team et al., 2024). We refer to this variant as **Full-Branch** in our experiments.

- **Last-layer fine-tuning.** As the lightest-weight alternative, we freeze all parameters except the final output layer of the branch network (**Last-Branch**), the final output layer of the trunk network (**Last-Trunk**), or both (**Last-Both**) (Xu et al., 2023; Zhu et al., 2023; Wu, 2024). By restricting updates to the output heads of each sub-network, this approach keeps the adaptation cost minimal while still allowing limited flexibility in both the coefficients and the basis functions.

These three strategies offer a clear trade-off between adaptation capacity and computational cost. We compare them empirically in Section 5.3, where we find that branch-only fine-tuning achieves accuracy comparable to full-network updates at a fraction of the cost, suggesting that the pretrained basis functions transfer well across tasks, also highlighting the practical feasibility of fast online adaption of NO controllers.

### 4.3 Meta-Training for Rapid Adaptation

The approaches taken in Section 4.2 are based on a neural operator that was pretrained with the standard behavioral cloning objective (5) and subsequently adapted to new tasks. As shown in Section 5, this pretrained operator generalizes well across a range of tasks and adaptation strategies. Here we investigate replacing the behavioral cloning loss with one that is designed for fast adaptation. This can have benefits in a number of settings, for instance when the amount of task-specific data available at deployment is much smaller than what was used during training. Additionally, when pretraining data itself is limited, meta-training can compensate: as we show in Section 5.4, SetONet-Meta (meta update branch only) improves over the pretrained model on P2P-Cost-Small, where standard pretraining alone does not fully capture the task distribution. To target these settings, we modify the training pipeline using a bi-level formulation inspired by MAML (Finn et al., 2017) that trains a neural operator whose initialization is specifically optimized for few-shot task adaptation. This procedure is split into *inner* and *outer* loops; we consider two variants that differ in which parameters participate in the inner loop, yielding different trade-offs between adaptation speed and representational flexibility. A commonly cited limitation of meta-learning is that the learned initialization may not generalize well to tasks far from the training distribution. We show that by adapting both the branch and trunk networks during the inner loop, SetONet-Meta-Full (meta update full network) is able to adapt to an OOD task on the Quadrotor environment.

**Bi-level training objective.** We adopt the episodic training structure of MAML but apply it to the neural operator architecture. Denote the full parameter set as $\theta = (\theta_{\text{branch}}, \theta_{\text{trunk}})$. In each training episode, we sample a batch of $B$ tasks $\{i_1, \ldots, i_B\}$ and, for each task $i$, split the available data into a *support* set $\mathcal{D}_i^{\text{tr}}$ used for inner-loop adaptation and a *query* set $\mathcal{D}_i^{\text{eval}}$ used for outer-loop evaluation. Both sets contain context data (sensor location–value pairs for the branch) and target state–control pairs (at query locations for the trunk and loss computation).

The inner loop performs a single gradient descent step on the support loss. We consider two choices for the scope of this update:

- **SetONet-Meta** (branch-only inner loop). Only the branch parameters $\theta_{\text{branch,i}}$ for task $i$ are updated in the inner loop while the trunk remains frozen:

$$\theta'_{\text{branch,i}} = \theta_{\text{branch}} - \alpha \nabla_{\theta_{\text{branch}}} \mathcal{L}_{\mathcal{D}_i^{\text{tr}}}(\theta_{\text{branch}}, \theta_{\text{trunk}}). \tag{12}$$

  This preserves the learned basis functions $\{b_k\}$ and restricts adaptation to finding new coefficients $\{c_k\}$ that best represent the target task's policy. The trunk parameters participate only in the outer loop, where they receive gradients that account for how well the basis functions support adaptation across the full task distribution. This strategy leverages the branch–trunk decomposition of the operator: the trunk learns basis functions that are optimized to be *reused* across tasks, while the branch learns an initialization from which a single gradient step yields a good task-specific policy.

- **SetONet-Meta-Full** (full-network inner loop). Both branch and trunk parameters $\theta_i$ for task $i$ are updated in the inner loop:

$$\theta'_i = \theta - \alpha \nabla_\theta \mathcal{L}_{\mathcal{D}_i^{\text{tr}}}(\theta), \tag{13}$$

  This allows the operator to adapt both the coefficients *and* the basis functions to a new task in a single step. The outer loop optimizes a shared initialization from which full-network adaptation is effective, analogous to standard MAML. This variant has strictly greater adaptation capacity than the branch-only strategy, since it can reshape the basis functions for each task, but it updates more parameters per step and may be more susceptible to overfitting when few demonstrations are available.

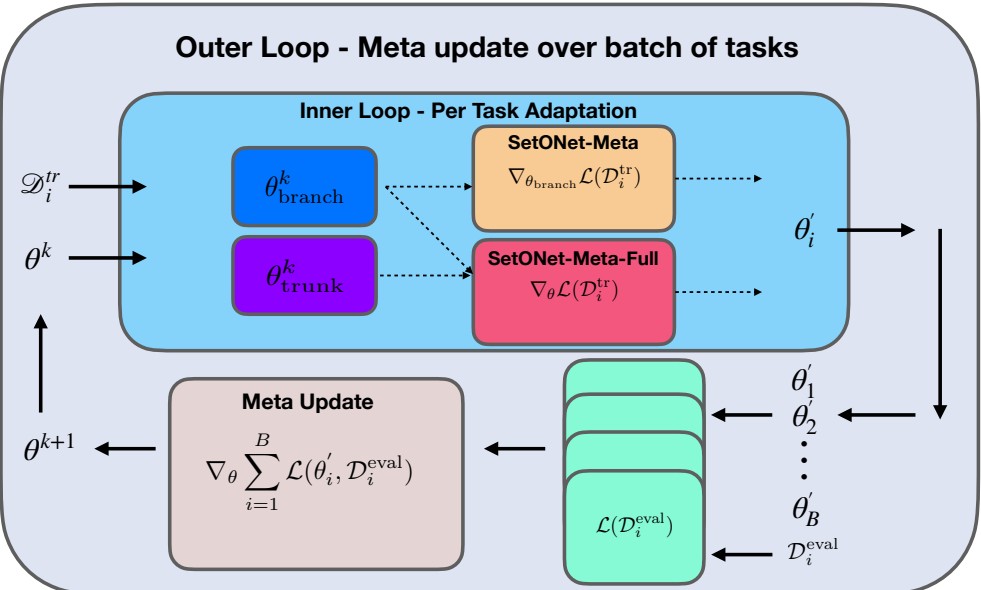

Figure 3: Overview of the meta-training procedure. The inner loop adapts the parameters $\theta^k$ to each task $i$ using the support set $\mathcal{D}_i^{\text{tr}}$. SetONet-Meta updates only the branch coefficients $\theta_{\text{branch}}$, while SetONet-Meta-Full updates all parameters $\theta$. The adapted parameters $\theta_i'$ from a batch of $B$ tasks are evaluated on held-out query sets $\mathcal{D}_i^{\text{eval}}$, and the outer loop computes the meta-gradient to update the shared initialization $\theta^{k+1}$.

In both cases, the outer loop updates *all* parameters, both $\theta_{\text{branch}}$ and $\theta_{\text{trunk}}$, by minimizing the post-adaptation loss on the query sets, averaged across the task batch:

$$\theta \leftarrow \theta - \beta \frac{1}{B} \nabla_\theta \sum_{i=1}^{B} \mathcal{L}_{\mathcal{D}_i^{\text{eval}}}(\theta'_{\text{branch},i}, \theta'_{\text{trunk},i}), \tag{14}$$

where $\beta$ is the outer learning rate. For SetONet-Meta, $\theta'_{\text{trunk},i} = \theta_{\text{trunk}}$ (unchanged by the inner loop), so the outer gradient with respect to $\theta_{\text{trunk}}$ flows only through the query loss evaluated at the adapted branch. For SetONet-Meta-Full, both components pass through the inner update via second-order differentiation. This architecture is shown in Figure 3 where we show the inner and outer training loops, as well as how the two proposed architectures fit in.

**Relationship to standard training and MAML.** Both meta-training variants occupy a middle ground between the standard pre-training pipeline of Section 3.3 and classical MAML. Like MAML, they use bi-level optimization to learn representations that are explicitly designed for rapid adaptation. SetONet-Meta-Full is closest to standard MAML in spirit, differing primarily in the architecture (a neural operator rather than a monolithic policy network); it learns a full-network initialization from which a single gradient step yields a good task-specific model. SetONet-Meta, by contrast, exploits the branch–trunk decomposition unique to neural operators: the trunk learns shared basis functions that remain fixed during adaptation, and the branch learns an initialization from which task-specific coefficients can be rapidly recovered. This restricted inner loop reduces the number of trainable parameters at adaptation time and lowers the risk of overfitting, at the cost of limiting representational flexibility to the span of the pretrained basis.

## 5 Experiments

We evaluate the neural operator model developed in Section 4 on four optimal control environments of increasing complexity and a higher-dimensional locomotion environment from the iMuJoCo benchmark (Section 5.1), where expert policies are trained via reinforcement learning. Our evaluation is organized into two

parts. First, in Section 5.2, we demonstrate that a neural operator can accurately approximate the solution operator across the task distribution and establish baseline accuracy across all environments (Table 2). We additionally examine how prediction accuracy varies with the number and ordering of context points provided at test time, including sizes not seen during training (i.e., *task resolution invariance*). Second, in Section 5.3, we evaluate the adaptation strategies of Sections 4.2 and 4.3: fine-tuning with expert demonstrations, a per-task comparison against MAML (Figure 6), adaptation using cost feedback on environments where expert data is unavailable or expensive to obtain, and a detailed study of the meta-trained operator on HalfCheetah-v3 that examines how adaptation performance scales with the number of demonstrations and gradient steps (Figure 10).

## 5.1 Environments

We consider four optimal control environments and a reinforcement learning environment from the iMuJoCo benchmark, summarized in Table 1 and further described below.

**Constructing the context set.** For each environment the branch network's context is built from simple sampling schemes of the cost and dynamics, and never requires solving the pOCP at deployment (Table 1, "Context"). We sample context points in one of three ways: for the cost-varying task (P2P-Cost) we draw states and controls uniformly over the workspace and evaluate the known stage cost; for the dynamics-varying tasks (P2P-Dynamics, Quadrotor, HalfCheetah) we sample one-step transitions $(\mathbf{x}, \mathbf{u}) \to \mathbf{x}'$ from short rollouts or logged interaction, which need only excite the dynamics rather than be expert-optimal; and for Obstacle the context is the observed obstacle field itself, one element per obstacle. Per-environment procedures are given below; the full sampling code is provided in the supplementary material. The number of context points sampled per iteration follows the resolution-invariance protocol of Section 5.2, where Figure 5 shows that roughly 16–32 samples suffice before error plateaus.

The **Point-to-Point Cost (P2P-Cost)** environment is a 2D point mass with linear dynamics and quadratic running and terminal costs, where each task corresponds to a different goal state $\mathbf{x_g}$ and the expert is a linear quadratic regulator (LQR) controller. The context set consists of state–control–cost tuples $\mathcal{C} = \{((x, u, t), \ell(x, u, t; \boldsymbol{\phi}))\}$ that encode the task-defining cost function. The stage and terminal costs are given by

$$\ell(\mathbf{x}_t, \mathbf{u}_t; \boldsymbol{\phi}) = (\mathbf{x}_t - \mathbf{x}_g)^\top Q (\mathbf{x}_t - \mathbf{x}_g) + \mathbf{u}_t^\top R \mathbf{u}_t, \qquad \ell_T(\mathbf{x}_T; \boldsymbol{\phi}) = (\mathbf{x}_T - \mathbf{x}_g)^\top Q_T (\mathbf{x}_T - \mathbf{x}_g), \qquad (15)$$

where $Q, Q_T \succeq 0$ and $R \succ 0$ are fixed weight matrices shared across all tasks and the task parameter $\boldsymbol{\phi} = \mathbf{x}_g$ specifies the goal location. The linear dynamics are given by

$$\mathbf{x}_{t+1} = A\mathbf{x}_t + B\mathbf{u}_t, \quad \text{where } A = \begin{bmatrix} 1 & 0 & \Delta t & 0 \\ 0 & 1 & 0 & \Delta t \\ 0 & 0 & 1 & 0 \\ 0 & 0 & 0 & 1 \end{bmatrix}, \ B = \begin{bmatrix} 0 & 0 \\ 0 & 0 \\ \Delta t & 0 \\ 0 & \Delta t \end{bmatrix}, \qquad (16)$$

where $\Delta t > 0$ is a fixed constant indicating an Euler time discretization step size. This is a standard LQR problem and a direct instance of the parametric optimal control formulation (1). In the experiments that follow, we include a P2P-Cost-Small that allows us to test how different models perform when there is less data to train.

**Point-to-Point Dynamics (P2P-Dynamics)** This environment uses the same point-mass state and control spaces as P2P-Cost but introduces parameter-dependent nonlinearities. The state is $\mathbf{x} = (p_x, p_y, v_x, v_y)$ and the control $\mathbf{u} = (a_x, a_y)$ is a commanded acceleration. The dynamics, discretized with an Euler step of size $\Delta t$, are

$$\mathbf{x}_{t+1} = \mathbf{x}_t + \Delta t \begin{bmatrix} v_{x,t} \\ v_{y,t} \\ \text{clip}(\mu\, a_{x,t}, -a^{\max}, a^{\max}) \\ \text{clip}(\mu\, a_{y,t}, -a^{\max}, a^{\max}) \end{bmatrix}, \qquad (17)$$

where $\text{clip}(x, a, b) = \max(a, \min(x, b))$. The control gain $\mu \in (0, 1]$ attenuates the commanded acceleration, and per-axis acceleration saturation enforces $\|\dot{v}\|_\infty \leq a^{\max}$. After each step, velocities are additionally clipped

Table 1: Summary of experimental environments. All OCP environments use the same SetONet architecture and training procedure; only the context and query definitions change across environments. "Context" lists the (location → value) pairs that form the branch network's per-task input ($\mathbf{x}'$ is the next state, $\ell$ the stage cost); how these are sampled is described in the per-environment paragraphs below and in the supplementary material. The iMuJoCo environment encodes tasks through logged state–action transitions. "Tasks" denotes the number of distinct task configurations and "Traj." the number of expert trajectories per task.

| Environment | $d_x/d_u$ | Expert | Tasks | Traj. | Task parameters | Context |
|---|---|---|---|---|---|---|
| P2P-Cost | 4/2 | LQR | 500 | 100 | $\mathbf{x}_g \in [-10, 10]^2$ | $(\mathbf{x}, \mathbf{u}) \to \ell$ |
| P2P-Cost-Small | 4/2 | LQR | 50 | 10 | $\mathbf{x}_g \in [-10, 10]^2$ | $(\mathbf{x}, \mathbf{u}) \to \ell$ |
| P2P-Dynamics | 4/2 | iLQR | 100 | 100 | $\mu, v^{\max}, a^{\max}$ | $(\mathbf{x}, \mathbf{u}) \to \mathbf{x}'$ |
| Quadrotor | 6/2 | iLQR | 100 | 20 | $m, L, \alpha$ | $(\mathbf{x}, \mathbf{u}) \to \mathbf{x}'$ |
| Obstacle | 4/2 | NLP | 500 | 60 | $n_{\mathrm{obs}}, (\mathbf{x}, \mathbf{y})_i$ | $(\mathbf{x}, \mathbf{y})_i \to r_i$ |
| HalfCheetah-v3 | 17/6 | SAC | | 53 | 100 | mass, limb, joints, friction | $(\mathbf{x}, \mathbf{u}) \to \mathbf{x}'$ |

to enforce $\|v\|_\infty \leq v^{\max}$. Each task is defined by a distinct dynamics parameterization $\boldsymbol{\psi} = (\mu, v^{\max}, a^{\max})$, making this an instance of equation 1 with fixed cost $\boldsymbol{\phi}$ and varying dynamics. The context set encodes evaluations of the dynamics function: $\mathcal{C} = \{((\mathbf{x}, \mathbf{u}), \mathbf{g}(\mathbf{x}, \mathbf{u}; \boldsymbol{\psi}))\}$. Because the clipping makes the dynamics nonlinear, expert trajectories are generated with iterative linear quadratic regulator (iLQR) (Tassa et al., 2012) rather than closed-form LQR.

**Planar Quadrotor** The planar quadrotor has a 6D state $\mathbf{x} = (y, z, \phi, \dot{y}, \dot{z}, \dot{\phi})$ and control $\mathbf{u} = (F_z, \tau)$ consisting of the total thrust and roll torque. The dynamics, discretized with an Euler step of size $\Delta t$, are

$$\mathbf{x}_{t+1} = \mathbf{x}_t + \Delta t \begin{bmatrix} \dot{y}_t \\ \dot{z}_t \\ \dot{\phi}_t \\ -\frac{1}{m} F_{z,t} \sin \phi_t \\ \frac{1}{m} F_{z,t} \cos \phi_t - g \\ \frac{1}{I} \tau_t \end{bmatrix}, \tag{18}$$

where $I = \alpha m L^2$ is the moment of inertia. Each task is defined by varying physical parameters $\boldsymbol{\psi} \in \mathbb{R}^3 = (m, L, \alpha)$: mass, arm length, and an inertia scaling factor. The context set encodes dynamics evaluations $\mathcal{C} = \{((\mathbf{x}, \mathbf{u}), \mathbf{g}(\mathbf{x}, \mathbf{u}; \boldsymbol{\psi}))\}$. Expert trajectories are computed with iLQR, targeting a fixed hover state.

**Obstacle Avoidance** This environment uses the same double-integrator dynamics as P2P-Cost (state $\mathbf{x} \in \mathbb{R}^4$, control $\mathbf{u} \in \mathbb{R}^2$, with $A$ and $B$ as defined above), but each task is defined by a randomly generated field of $n_{\mathrm{obs}} \in [2, 6]$ circular obstacles, each parameterized by its center $(\mathbf{x}, \mathbf{y})_i$. The expert solves a constrained nonlinear programming problem (NLP) via IPOPT (Wächter & Biegler, 2006; Andersson et al., 2019), minimizing control effort subject to hard collision avoidance constraints. Unlike the other environments, the task-defining function is not directly available as a smooth cost; instead, the context set encodes the obstacle geometry as a collection of tuples $\mathcal{C} = \{(\mathbf{x}_i, \mathbf{y}_i, r_i)\}_{i=1}^{n_{\mathrm{obs}}}$, which serves as a finite-dimensional proxy for the underlying constraint structure. This is an instance of (1) with fixed dynamics, where the task variation enters through the constraint structure rather than the cost function directly.

**iMuJoCo** To test generalization beyond the model-based optimal control problems, we additionally evaluate on HalfCheetah-v3 from the iMuJoCo benchmark (Patacchiola et al., 2023), which provides families of MuJoCo agents whose physical parameters: body mass, limb length, joint range, and surface friction, are systematically varied from a default configuration. HalfCheetah-v3 ($d_x = 17$, $d_u = 6$, 53 configurations) is a high-dimensional locomotion task where expert policies are trained via Soft Actor-Critic (SAC) (Haarnoja et al., 2018) rather than computed from a known dynamics model, and the resulting demonstrations are collected as offline rollouts.

All of our experiments are done on a single GPU (NVIDIA GeForce RTX 5070). They all use the same SetONet architecture and training procedure. The branch network uses a set encoder consisting of a multilayer

perceptron (MLP) $\phi$ applied to each element of the context, followed by mean-pooling aggregation and a post-aggregation MLP $\rho$; the trunk network is a standard MLP. The full list of hyperparameters and architecture is included in the supplementary material. All models are trained using the Adam optimizer (Kingma & Ba, 2014) with a learning rate of $\eta = 10^{-3}$ to minimize the mean-squared-error loss (9). During training, the number of context points $m$ is sampled uniformly from a predefined set at each iteration, encouraging the model to perform well across varying context resolutions. For each environment, the task dataset is split 80%/20% into training and held-out test tasks; all reported metrics are computed on held-out tasks not seen during training. States and context values are normalized to zero mean and unit variance using statistics computed from the training set. Models are implemented in JAX using the Equinox library (Kidger & Garcia, 2021).

## 5.2 Neural Operator Fitting and Task Resolution

We first evaluate the ability of the SetONet model to approximate the solution operator across each task distribution when trained with the behavioral cloning objective (5). Rather than reporting raw mean squared error (MSE), we use the relative $L^2$ error

$$\text{Relative } L^2 \text{ error} = \frac{\|\mathcal{T}_\theta(\ell_i)(\mathbf{y}) - \mathbf{u}^*\|_2}{\|\mathbf{u}^*\|_2}, \tag{19}$$

which normalizes by the magnitude of the expert control signal and provides an interpretable, scale-consistent metric across environments. To assess generalization, we report this metric on held-out tasks not seen during training, and additionally perform model rollouts to test how well the learned policy performs on states that differ from those visited by the expert. We then examine *task resolution invariance*: the sensitivity of the operator's predictions to the number and ordering of context points provided at test time, including context set sizes not seen during training.

**Fitting the operator.** For each environment, we train a SetONet on a distribution of tasks and evaluate on held-out tasks not seen during training. At test time, the trained operator receives a context set from the new task and predicts the corresponding feedback policy. We report results under two evaluation settings: *expert-states*, in which the model predicts controls at the same states visited by the expert, and *model-rollout*, in which the predicted policy is executed from the same initial condition without any expert feedback. The model-rollout setting is the more demanding test, as it exposes the model to compounding errors—a well-known challenge in imitation learning often referred to as *distributional shift* (Ross et al., 2011), where small prediction errors accumulate and push the state trajectory away from the expert's distribution.

To quantify expert-states accuracy, we compute the behavioral cloning loss (5) across all expert trajectories for each task and report the relative $L^2$ error (19). The zero-shot rows of Table 2 (0 steps, Pretrained) report these errors for all four OCP environments and P2P-Cost-small. The pretrained operator achieves low relative error on P2P-Cost and Quadrotor, where the task variation is well-covered by the training distribution. P2P-Cost required more demonstration trajectories per task to achieve comparable error, which we attribute to the wide range of goal locations ($\mathbf{x}_g \in [-10, 10]^2$): distant goals produce larger control signals and correspondingly noisier relative error. The dynamics-varying (P2P-Dynamics) and obstacle environments show higher pretrained error, reflecting the greater diversity of these task distributions; these are the settings where adaptation yields the largest gains, as detailed in Section 5.3.

Figure 4 provides a qualitative view of these results across three environments, overlaying expert-states predictions and model-rollout controls on the expert for representative held-out tasks. Across all three environments, the learned operator produces model-rollout trajectories that closely follow the expert without any online correction. The Planar Quadrotor presents the most challenging fitting problem due to its higher-dimensional state space and coupled dynamics, yet the model still captures the qualitative control profile and the rollouts reach the correct targets despite minor transient deviations.

**Task resolution invariance.** A key property of the set-based context representation is that the learned operator can be evaluated with context sets of varying cardinality at test time, without retraining. This *task*

Table 2: Adaptation performance across four OCP environments and a reduced-data variant (P2P-Small), reported as relative $L^2$ error (mean $\pm$ std over 5 seeds), using a single expert demonstration for adaptation and 32 trajectories for evaluation. The top block reports zero-shot predictions with no gradient updates; the lower blocks report performance after 1 and 25 gradient steps. Bold indicates the best (lowest) error within each step block per column. "–" indicates divergence. All errors are relative $L^2$ (Eq. 19).

| Method | P2P-Cost | P2P-Small | P2P-Dyn. | Quadrotor | Obstacle |
|---|---|---|---|---|---|
| *0 steps (zero-shot)* | | | | | |
| Pretrained | **.048** $\pm$ .001 | .101 $\pm$ .009 | **.179** $\pm$ .012 | **.063** $\pm$ .005 | **.238** $\pm$ .001 |
| MAML | .583 $\pm$ .018 | .920 $\pm$ .016 | .802 $\pm$ .006 | .549 $\pm$ .092 | 1.000 $\pm$ .000 |
| SetONet-Meta | .075 $\pm$ .002 | **.084** $\pm$ .006 | .276 $\pm$ .033 | .185 $\pm$ .022 | .287 $\pm$ .001 |
| SetONet-Meta-Full | .118 $\pm$ .009 | .130 $\pm$ .010 | .195 $\pm$ .033 | .148 $\pm$ .022 | .395 $\pm$ .000 |
| *1 gradient step* | | | | | |
| SetONet-FT | .080 $\pm$ .002 | .090 $\pm$ .002 | .238 $\pm$ .016 | .069 $\pm$ .004 | **.232** $\pm$ .001 |
| Last-Branch | **.045** $\pm$ .001 | .095 $\pm$ .009 | .180 $\pm$ .012 | .061 $\pm$ .005 | .238 $\pm$ .001 |
| Last-Both | .047 $\pm$ .001 | .088 $\pm$ .007 | .182 $\pm$ .012 | **.059** $\pm$ .005 | .237 $\pm$ .001 |
| MAML | .581 $\pm$ .019 | .917 $\pm$ .019 | .898 $\pm$ .024 | .121 $\pm$ .008 | 1.000 $\pm$ .000 |
| SetONet-Meta | .074 $\pm$ .003 | **.083** $\pm$ .007 | .255 $\pm$ .029 | .170 $\pm$ .020 | .287 $\pm$ .001 |
| SetONet-Meta-Full | .117 $\pm$ .009 | .129 $\pm$ .009 | **.099** $\pm$ .010 | .066 $\pm$ .005 | .311 $\pm$ .001 |
| *25 gradient steps* | | | | | |
| SetONet-FT | .065 $\pm$ .001 | .090 $\pm$ .002 | .143 $\pm$ .009 | .065 $\pm$ .007 | .237 $\pm$ .001 |
| Last-Branch | **.044** $\pm$ .001 | **.075** $\pm$ .004 | .177 $\pm$ .010 | .059 $\pm$ .004 | **.231** $\pm$ .001 |
| Last-Both | .048 $\pm$ .001 | .076 $\pm$ .003 | .171 $\pm$ .012 | **.057** $\pm$ .005 | .234 $\pm$ .001 |
| MAML | .569 $\pm$ .017 | .898 $\pm$ .013 | – | .118 $\pm$ .008 | 1.000 $\pm$ .000 |
| SetONet-Meta | .069 $\pm$ .002 | .077 $\pm$ .006 | .094 $\pm$ .009 | .071 $\pm$ .006 | .285 $\pm$ .001 |
| SetONet-Meta-Full | .096 $\pm$ .005 | .102 $\pm$ .008 | **.081** $\pm$ .007 | .073 $\pm$ .004 | .306 $\pm$ .001 |

*resolution invariance* arises because the branch network learns a set-based representation of the task from the context set, rather than relying on a fixed ordered discretization. The permutation-invariant aggregation ensures that this representation is well defined for variable-sized, unordered context sets. This is particularly appealing in control settings, where the amount of expert data may vary across tasks (e.g., in the number of trajectories or rollout lengths), allowing the model to flexibly leverage available data at inference time without retraining.

Figure 5 evaluates this property across all four environments. In each panel, the $x$-axis varies the number of context samples provided to the branch encoder at test time, including sizes both seen and not seen during training. Lines show the median relative $L^2$ error over held-out tasks and shaded regions indicate the interquartile range (IQR). We report median and IQR rather than mean and standard deviation because the error distribution across tasks is right-skewed: a small number of particularly challenging configurations can produce large errors that disproportionately inflate the mean.

For P2P-Cost, P2P-Dynamics, and the Planar Quadrotor, all three environments exhibit a consistent pattern: error is noticeably higher and more variable with very few context points (1–4), but decreases steadily and stabilizes by around 16–32 samples. Beyond this point, performance remains steady across the full range, including at context sizes not seen during training. This indicates that the branch encoder requires only a modest number of samples to extract reliable task information, and does not overfit to a particular context cardinality. In the dynamics-varying environments (P2P-Dynamics, Quadrotor), the initial decrease is expected: a single observation of $\mathbf{x}_{t+1} = \mathbf{g}(\mathbf{x}_t, \mathbf{u}_t; \boldsymbol{\psi})$ provides limited information about the dynamics parameters, but a handful of samples suffices to approximately identify them. P2P-Cost exhibits a similar pattern, with error dropping sharply from 1 to 8 context points before leveling off.

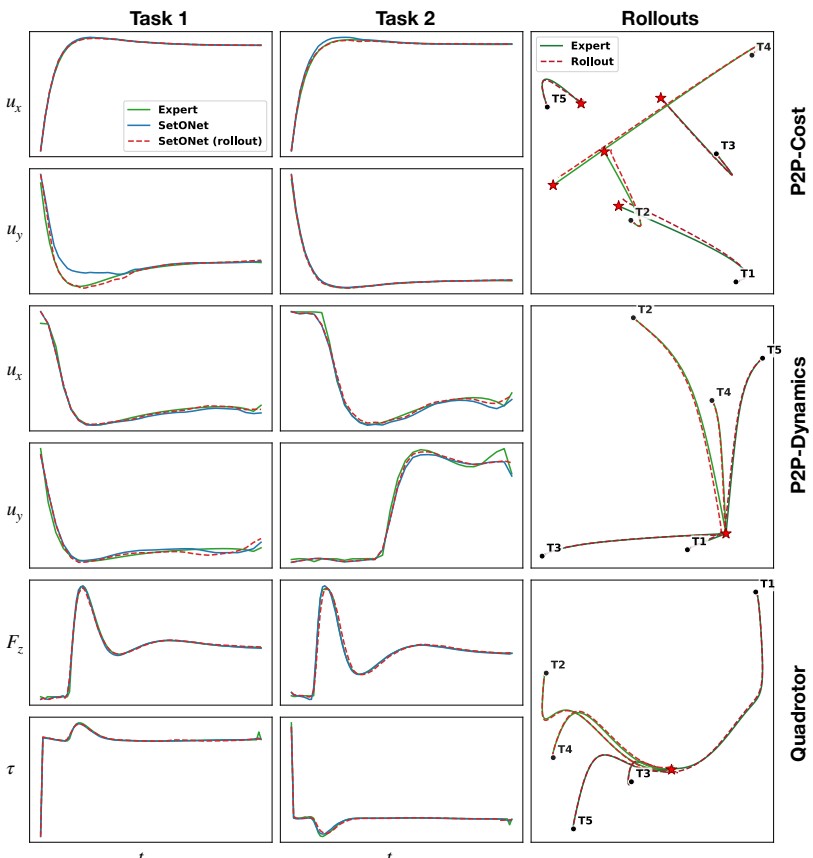

Figure 4: Operator fitting results across three environments. Each row group shows two control dimensions for a given environment, with columns displaying predictions on two representative tasks. We then show the corresponding state-space rollouts for the two tasks (T1, T2) along with 3 more (T3-T5). Solid green lines denote expert demonstrations, solid blue are SetONet predictions at the expert state locations, and dashed lines denote model-rollouts using the learned operator policy.

The Obstacle Avoidance environment (Figure 5, bottom right) presents a qualitatively different picture. Unlike the other three environments, the median error increases with the number of obstacles. We attribute this to the inherent growth in task complexity: configurations with more obstacles create tighter passages and more constrained feasible trajectories, making the mapping from obstacle configuration to collision-free policy harder to approximate. The widening IQR at higher obstacle counts further suggests that some configurations are significantly harder than others, likely those with narrow corridors or near-degenerate passages. Notably, the model maintains reasonable accuracy at obstacle counts not seen during training (3 and 5 obstacles, red ticks), demonstrating that the set-based representation interpolates smoothly between the training configurations ($n_{\mathrm{obs}} \in \{2, 4, 6\}$).

This resolution invariance is a practical advantage: at deployment, the operator can produce reasonable policies from whatever context data is available, whether more or fewer samples than were used during training.

## 5.3 Task-Specific Adaptation

We now evaluate the adaptation strategies introduced in Section 4.2 on the held-out tasks not seen during training. This section focuses on fine-tuning using expert demonstrations (compared against a MAML baseline) and adaptation using cost feedback. Meta-trained operators are evaluated separately in Section 5.4. A central objective of this evaluation is to provide a systematic comparison between neural operator adaptation

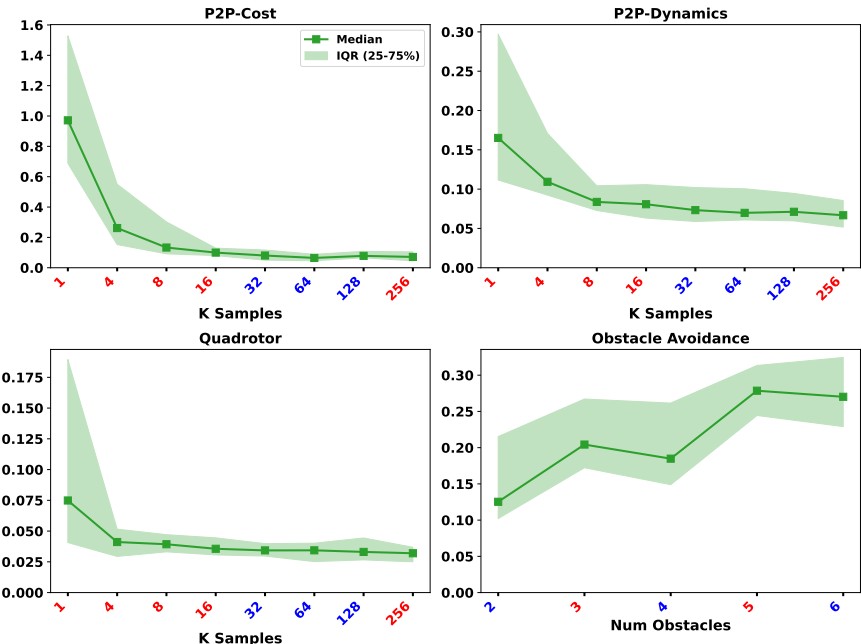

Figure 5: Task resolution invariance across all four control environments. Lines show median relative $L^2$ error over held-out tasks; shaded regions indicate the inter-quartile range. Blue tick marks denote context sizes seen during training; red tick marks denote sizes not seen during training. The $x$-axis for the first three environments is the number of context samples; for Obstacle Avoidance it is the number of obstacles. The model was trained on obstacle configurations with 2, 4, and 6 obstacles.

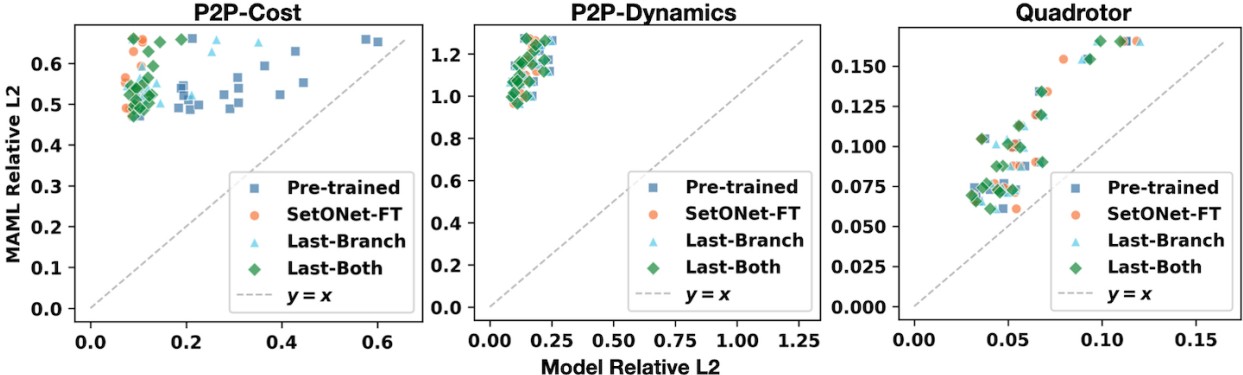

Figure 6: Per-task comparison of MAML against four SetONet-based methods across three OCP environments after 25 gradient steps using 10 expert trajectories for adapting and evaluation. Each point represents a single held-out task, with the $x$-axis showing the method's relative $L^2$ error and the $y$-axis showing MAML's error on the same task. Points above the diagonal ($y = x$) indicate tasks where MAML performs worse. MAML is consistently dominated across nearly all tasks in P2P-Cost and P2P-Dynamics, while in the Quadrotor environment the gap narrows, with some tasks falling near the diagonal.

and meta-learning approaches popular in control and reinforcement learning, such as MAML. Through these comparisons, we demonstrate that efficient fine-tuning of neural operators is competitive, and often superior to MAML, without incurring additional meta-training cost.

**Adaptation with expert demonstrations.** Table 2 reports the mean relative $L^2$ error across five OCP environments after 0, 1, and 25 gradient steps of adaptation using a single expert demonstration. Among the non-meta-trained methods, Last-Branch and Last-Both, which update only the last layer of the branch

network or the last layer of both networks, respectively, consistently perform well: a single gradient step is often sufficient to match or improve upon the pretrained operator, and 25 steps yield further gains on the dynamics-varying environments (P2P-Dynamics, Quadrotor). SetONet-FT, which updates the full network, shows the largest improvements on P2P-Dynamics, reducing error from .179 to .143 after 25 steps, but provides little benefit on P2P-Cost and Obstacle where the pretrained model is already accurate.

MAML, by contrast, produces high error across all environments and step counts, often failing to improve over its poor zero-shot initialization. Figure 6 provides a per-task view of this gap: each point compares MAML's error against one of the four SetONet-based methods on the same held-out task. In P2P-Cost and P2P-Dynamics, nearly all points lie above the diagonal, indicating that MAML is dominated on virtually every individual task. In the Quadrotor environment the margin narrows, with some tasks falling near the diagonal, though the SetONet methods still hold an overall advantage.

**Adaptation with cost feedback.** In some settings, expert demonstrations for the target task may be unavailable, the expert may have been trained on a cost function that we would like to modify, or only suboptimal policies may be accessible. Cost-based fine-tuning addresses these cases by adapting the operator directly on a downstream cost function, assuming knowledge of both the cost and environment dynamics. The operator is adapted by differentiating through model-rollouts, as described in Section 4.2.

Figure 7a illustrates this on out-of-distribution P2P-Cost tasks. The surrogate is the same LQR objective used during training ((15)), differentiated end-to-end through the known linear dynamics. After fine-tuning, all three strategies (full network, branch-only, and last-layer ) converge to costs comparable to the expert, demonstrating that the operator can be adapted purely from the task objective without any expert data.

Figure 7b applies cost-based fine-tuning to held-out obstacle avoidance tasks. Here the surrogate cost combines a soft collision penalty, $w_{\text{coll}} \sum_i \exp\!\big(-\alpha(d_{i,t} - r_i - m)\big)$, where $d_{i,t}$ is the distance from the agent to obstacle $i$, $r_i$ is the obstacle radius, and $m = 0.2$ is a safety margin, with a control regularization term $w_{\text{ctrl}} \|u_t\|^2$ and a terminal goal-reaching penalty, evaluated along a differentiable rollout through double-integrator dynamics ($w_{\text{coll}} = 10$, $w_{\text{ctrl}} = \Delta t$, $\alpha = 15$). The pretrained operator is trained purely by behavioral cloning of the expert controls and never observes the collision constraint or cost during pretraining; constraint information enters only through the surrogate cost at fine-tuning time. The pretrained SetONet spent an average of $\sim 3.5 \times$ the number of collision timesteps as the expert. All three fine-tuning strategies reduce collision time well below the expert while successfully reaching the goal on every task. Because the expert solver enforces collision avoidance through hard constraints rather than directly minimizing collision time, the fine-tuned models can achieve fewer collision timesteps by explicitly penalizing proximity to obstacles in the surrogate cost. This high collision count for the pretrained model reflects the nature of behavioral cloning rather than task difficulty: the imitation objective contains no feasibility term, and small control errors compound in closed loop (Ross et al., 2011), so trajectories that graze a zero-margin, boundary-hugging expert cross into collision. Together, these two experiments demonstrate that cost-based adaptation can refine the operator directly from the task objective, without requiring any expert demonstrations.

## 5.4 Meta-Trained Operator

The fine-tuning results above rely on a model that was pretrained with the standard behavioral cloning objective. We now evaluate whether the meta-training procedure of Section 4.3, which explicitly optimizes for rapid post-adaptation performance, can improve adaptation efficiency.

Returning to Table 2, the zero-shot rows reveal a clear difference between the two meta-trained variants. SetONet-Meta achieves the lowest zero-shot error on P2P-Small (.084) and competitive error on P2P-Cost (.075), outperforming SetONet-Meta-Full in both cases. On P2P-Dynamics, however, SetONet-Meta-Full produces the best zero-shot result (.195), suggesting that the ability to adapt basis functions, not just coefficients, matters when the task variation is more complex. After 25 gradient steps, SetONet-Meta-Full achieves the lowest error on P2P-Dynamics (.081), while SetONet-Meta remains competitive on P2P-Small (.077). Both meta-trained variants underperform Last-Branch and Last-Both on P2P-Cost and Quadrotor, where the pretrained operator is already accurate and the additional meta-training offers little benefit.

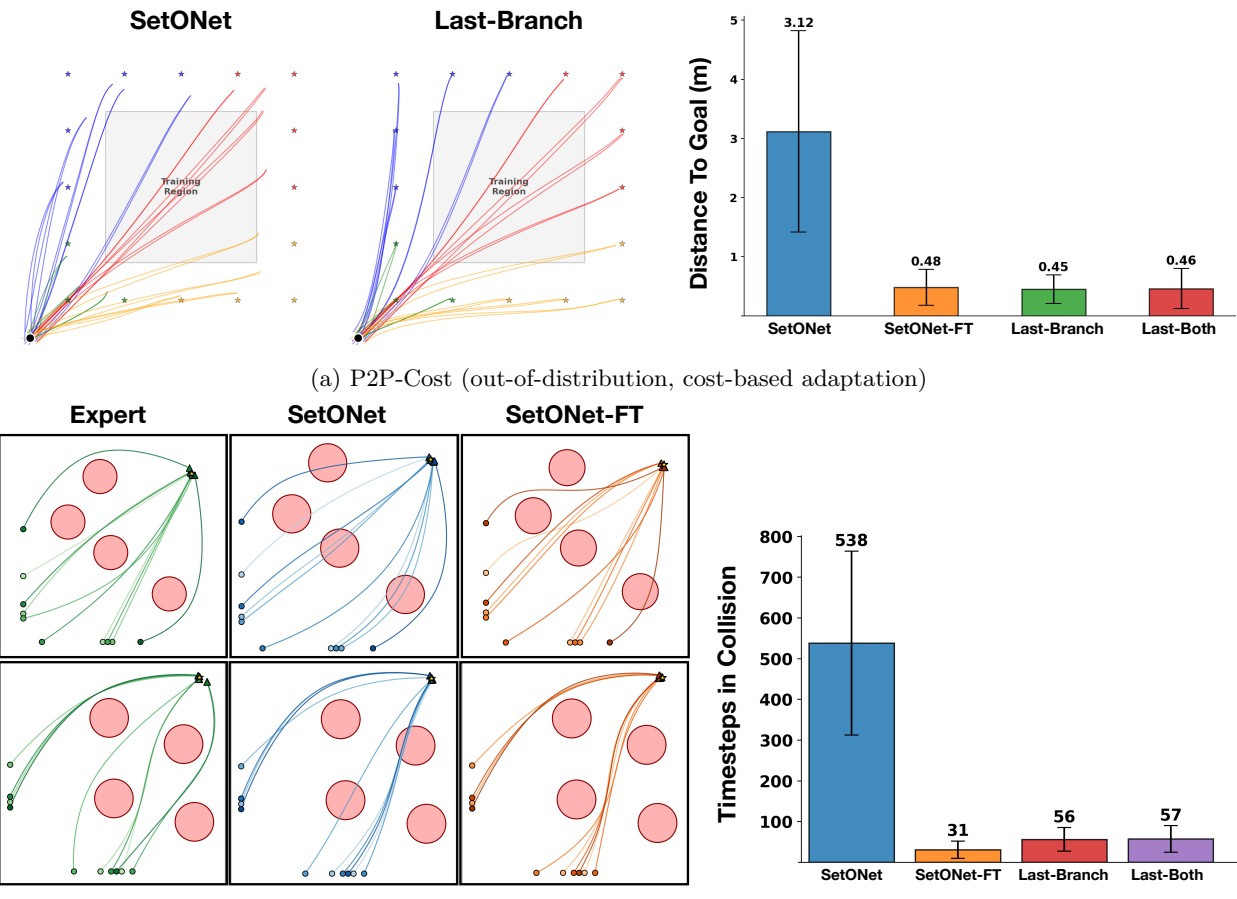

(a) P2P-Cost (out-of-distribution, cost-based adaptation)

(b) Obstacle Avoidance (held-out tasks, cost-based adaptation)

Figure 7: Cost-based fine-tuning across two environments. **(a)** Out-of-distribution P2P-Cost tasks with both goal locations and start position outside the training region (gray). The distance is $L^2$ in meters. The pretrained SetONet fails to reach distant goals (mean terminal distance of 3.12). After adaptation, all methods recover accurate trajectories, with Last-Branch (0.45) slightly outperforming SetONet-FT (0.48), indicating that the pretrained basis functions (trunk network) generalize well and only the coefficients require updating. **(b)** Held-out obstacle avoidance tasks. All cost-based fine-tuning strategies reduce collision time well below the pretrained model. Collision time is measured per trajectory as the total number of in-collision timesteps, averaged over held-out tasks with 2, 4, and 6 obstacles.

We define an out-of-distribution (OOD) task as one whose task parameters lie outside the convex hull of the parameters seen during pretraining, so that the target policy need not lie within the span of the pretrained basis. We evaluate this on a Quadrotor task in which the goal location is shifted outside the training region. Figure 8 compares all adaptation methods on this task. The pretrained SetONet as expected exhibits the largest error. SetONet-FT and SetONet-Meta-Full achieve the lowest errors, demonstrating the importance of trunk adaptation on an OOD task. We see that the meta-trained initialization enables effective adaptation even when the target task lies outside the training distribution. This is a setting where the pretrained basis functions do not generalize well, and adapting both coefficients and basis functions provides a clear advantage.

**Meta-training on HalfCheetah-v3.** We next evaluate the meta-trained operators on HalfCheetah-v3, where expert policies are trained via reinforcement learning (SAC). Figure 9 shows the first three control dimensions for representative held-out HalfCheetah-v3 configurations. Unlike the smooth, low-frequency control signals of the OCP environments, the expert policies here produce high-frequency, multi-dimensional

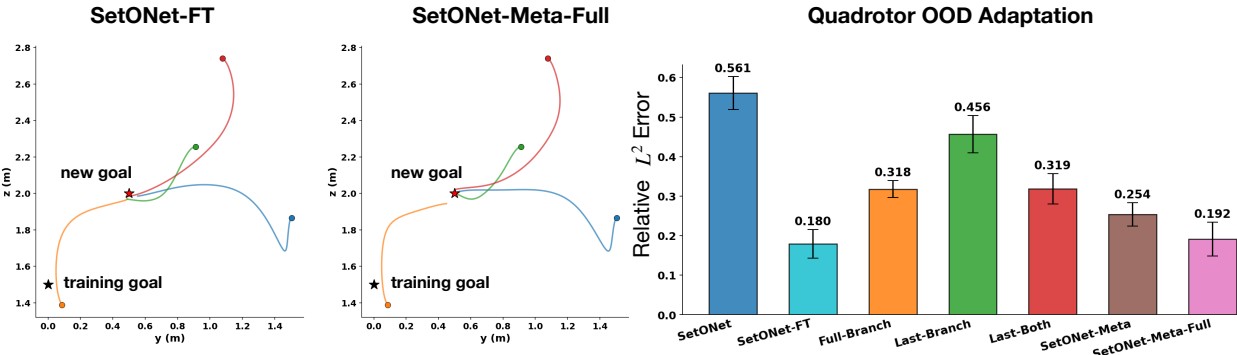

Figure 8: Out-of-distribution fine tuning on a Quadrotor task. Left two panels show multiple representative trajectories for SetONet-FT and SetONet-Meta-Full adapting to a shifted goal location. The bar plot compares relative $L^2$ error across all fine-tuning and meta-learning methods.

outputs that the operator must approximate from noisy, suboptimal demonstrations, making accurate few-shot adaptation considerably more demanding.

Figure 10 examines these trade-offs in greater depth, comparing four adaptation strategies across varying numbers of expert demonstrations (panels) and gradient steps (horizontal axis). Several patterns emerge. First, MAML and SetONet-Meta-Full consistently outperform SetONet-FT in the low-data regime (1 and 5 demonstrations), achieving lower error across all gradient step budgets. Since both MAML and SetONet-FT perform full-network adaptation, this advantage is attributable to the meta-trained initialization rather than the scope of parameters being updated: the bi-level training objective produces a starting point from which a single gradient step already yields a strong task-specific model. Second, SetONet-FT benefits most from additional demonstrations and gradient steps, closing the gap with the meta-trained methods at 10 demonstrations and overtaking them at 25 demonstrations with 50 or more gradient steps. This is expected—with sufficient data, the quality of the initialization matters less, and SetONet-FT is free to converge to any solution without constraints imposed by the inner-loop architecture chosen at meta-training time. Third, SetONet-Meta plateaus early, confirming that restricting the inner loop to the branch network alone limits adaptation capacity in this higher-dimensional environment.

These results highlight a practical trade-off: when the adaptation budget is limited to a small support set and few gradient steps, the meta-trained initialization provides a significant advantage over standard pretraining. As more data and compute become available, standard fine-tuning catches up and eventually achieves the lowest error. The choice between strategies thus depends on the deployment setting: meta-training is preferable for rapid, low-resource adaptation, while standard fine-tuning is preferable when a larger adaptation budget is available. Practical choices can be made problem-specific, in cases where rapid task switching is required during online deployment, meta-training of the neural operator controller can provide a significant boost in efficiency.

## 6 Limitations

We note three limitations of our approach: reliance on expert quality, the assumptions required for cost-based adaptation, and the scope of our out-of-distribution results.

**Expert quality.** Because the operator is trained by behavioral cloning, the quality of its basis functions is bounded by the experts used for pretraining, as with any imitation-learning method. Unbiased demonstration noise is largely averaged out by the shared trunk basis, but systematic expert bias is inherited and cannot be corrected by added capacity. In our experiments this is mild: the classical solvers (LQR, iLQR, NLP) are at or near optimal, and only the SAC expert for HalfCheetah-v3 is genuinely suboptimal. Where a differentiable model is available, cost-based adaptation can exceed a suboptimal expert by optimizing the

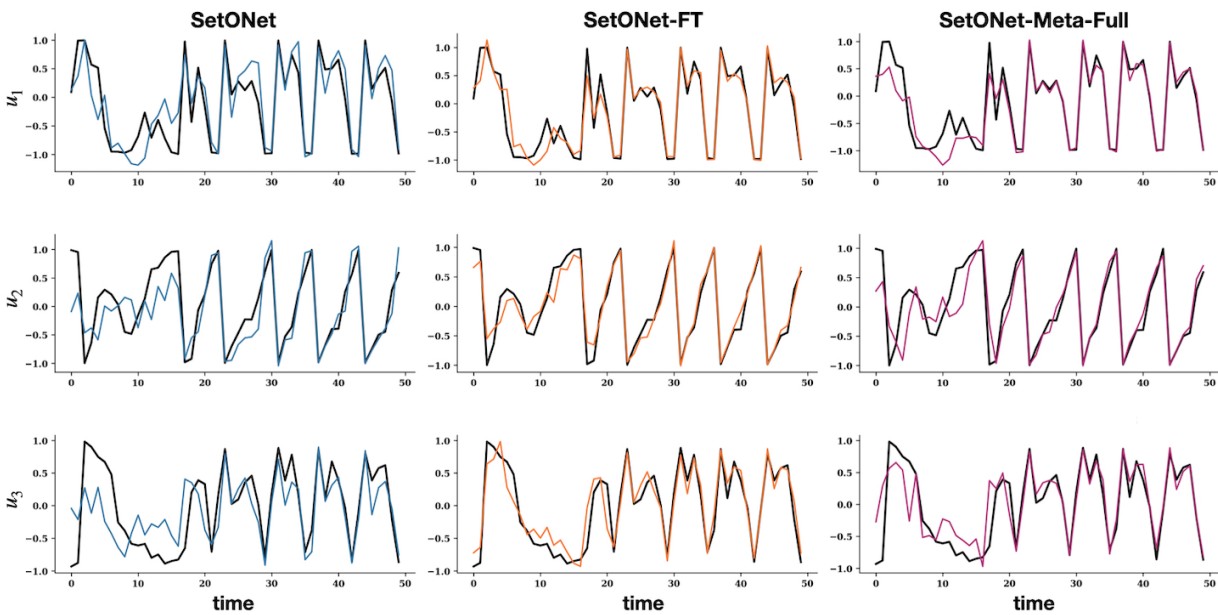

Figure 9: Control predictions on a held-out HalfCheetah-v3 task, showing the first three control dimensions ($u_1, u_2, u_3$, rows) for three methods (columns) over time. Each panel compares the method's prediction (colored) against the expert (black) after 25 gradient steps using 10 expert trajectory demonstrations of 100 timesteps each. SetONet-FT most closely tracks the expert across all dimensions.

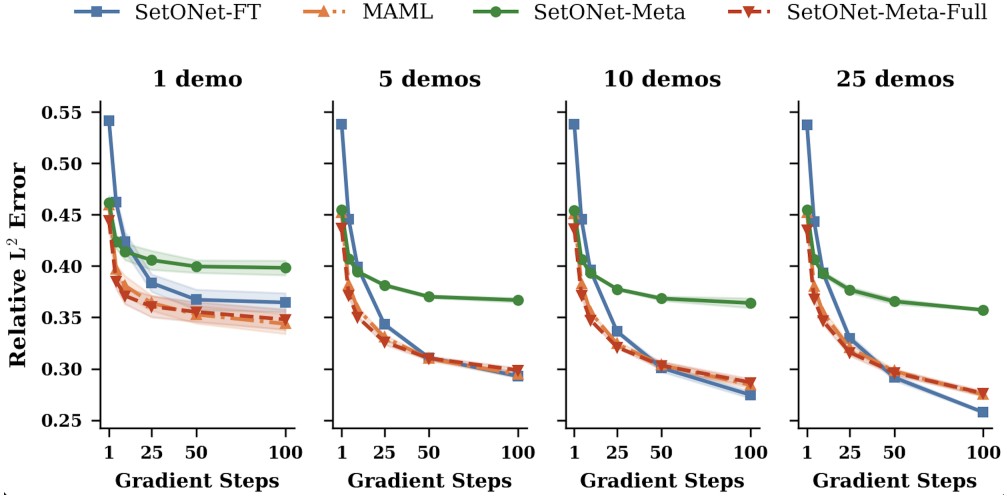

Figure 10: Adaptation performance on held-out HalfCheetah-v3 tasks as a function of the number of expert demonstrations (panels) and gradient steps (horizontal axis). Lines show mean relative $L^2$ error over 5 seeds; shaded regions indicate $\pm 1$ standard deviation. With few gradient steps (1–5), the meta-trained variants outperform SetONet-FT, particularly in the low-data regime (1 and 5 demos). As the adaptation budget increases, MAML and SetONet-Meta-Full improve steadily while SetONet-FT has the largest decrease in error as the number of available samples is increased.

task objective directly (as in our Obstacle results); otherwise, surpassing the expert would require interactive or offline-to-online refinement (Ross et al., 2011; Ball et al., 2023).

**Assumptions of cost-based adaptation.** Unlike the operator's zero-shot and demonstration-based paths, which need only pointwise context, cost-based fine-tuning requires the trajectory cost to be differentiable with respect to the policy parameters, i.e. a differentiable dynamics model and cost, as in (11). This is weaker than a closed-form model (any differentiable computation—integrator, simulator, or learned model—suffices) but stronger than black-box access. It extends to learned dynamics (subject to model error) and to reparameterizable stochastic dynamics, though we evaluate only the deterministic, known-dynamics case; differentiating through long rollouts can also be ill-conditioned (Metz et al., 2021), which we mitigate with gradient clipping.

**Scope of out-of-distribution claims.** Our out-of-distribution evidence is a single shift type: a goal placed outside the training region on the Quadrotor (Figure 8). Such a shift can move the target policy outside the span of the learned basis, which is why trunk adaptation, not coefficient adaptation alone, is required to recover it. We do not claim general robustness to arbitrary shifts. Whether trunk adaptation recovers an out-of-span target is open: the universal approximation property (Chen & Chen, 1995) guarantees suitable parameters *exist* given enough capacity and data, but not that finite-step adaptation reaches them. Harder shifts, such as dynamics far outside the training range, are left to future work.

## 7 Conclusion

We have presented a general framework for multi-task optimal control based on neural operators, demonstrating that the operator learning perspective offers a principled and practical approach to learning control policies across families of related tasks. By modeling the mapping from task-defining functions to optimal feedback policies, neural operators naturally capture the shared structure underlying parametric control problems while remaining flexible enough to accommodate diverse task variations.

Here we highlight a number of key results from this work. First, we showed through various experiments how the SetONet model can closely approximate the solution operators defined in equation 4. We demonstrated different forms of generalization on held-out tasks, including model rollouts that closely matched the expert and task resolution invariance, a property that is particularly valuable when the amount of available task information varies at deployment. Second, the branch–trunk decomposition enables effective task-specific adaptation, and we presented a number of adaptive model choices that target different parts of the network. We discussed the trade-offs of each, which depend on the available training data and its relationship to the downstream task. Third, we introduced two meta-training variants, SetONet-Meta and SetONet-Meta-Full, which explicitly optimize the initialization for rapid few-shot adaptation. SetONet-Meta provides data-efficient updates by restricting adaptation to the branch coefficients, while SetONet-Meta-Full adapts both coefficients and basis functions, yielding the largest gains on out-of-distribution tasks such as P2P-Dynamics and HalfCheetah-v3. Both consistently outperform MAML, which struggles to adapt in most environments. Finally, cost-based adaptation demonstrated that the operator can be refined directly from the task objective without any expert demonstrations, using only the cost function and the dynamics model.

Cost-based adaptation opens several directions for further exploration. For example, online data collection strategies such as DAgger (Ross et al., 2011) could iteratively refine the operator by querying the expert only in states actually visited under the learned policy, improving closed-loop performance while limiting the total number of expert solves. More broadly, although we explored several strategies for adaptation and fine-tuning, many remain unexplored, including methods designed specifically for out-of-distribution settings (Kumar et al., 2022). In this work, we restricted attention to settings in which a single input function varies across tasks (either cost or dynamics). Extending the operator learning framework to problems where tasks are defined by simultaneous variation in multiple function spaces is an open and practically relevant direction: for instance, in the point-to-point environments, one could vary both the cost function and the dynamics simultaneously. Recent work on meta-learning with neural operators (Wang et al., 2024a) takes a *multi-operator* approach to this problem, but focuses on a single application domain and does not provide theoretical analysis, leaving the general setting largely unexplored.

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

## A  Architectures and Hyperparameters

This section summarizes the model architectures and hyperparameters used in all experiments. To avoid repetition, values that are shared across environments are collected once in Table 3; the subsequent tables list only the quantities that vary by environment. All values are also provided in machine-readable form (`configuration_overview.md` and the `configs/` directory) in the supplementary code.

**SetONet architecture.** The branch is a DeepOSet set encoder: a per-element MLP $\phi$ is applied to each context element, the results are combined by attention-pooling aggregation, and a post-aggregation MLP $\rho$ produces the $p$ task-dependent coefficients $\{c_k\}$. The trunk is an MLP mapping a query $\mathbf{y} = (\mathbf{x}, t)$ to $p \times$ output_dim basis values, and the predicted control is $\hat{\pi}(\mathbf{y}) = \sum_{k=1}^{p} c_k \, b_k(\mathbf{y}) + $ bias. With the MLP widths fixed at 128 (Table 3), three architectural quantities vary across environments—the number of basis functions $p$, the depth (number of hidden layers, equal in $\phi$, $\rho$, and the trunk), and the number of attention tokens—as listed in Table 4. The P2P-Dynamics operator additionally applies min–max input normalization inside the model.

Table 3: Default settings, shared across all environments unless a later table notes an exception.

| Item | Value |
|------|-------|
| Operator architecture | SetONet (branch–trunk; attention-pooled, permutation-invariant set-encoder branch) |
| Framework | JAX + Equinox |
| MLP width (all of $\phi$, $\rho$, trunk, and the MAML-baseline MLP) | 128 |
| Branch aggregation | Multi-head attention pooling, $n_{\text{heads}} = 4$, $n_{\text{tokens}} = 4$ *(P2P-Dynamics: $n_{tokens} = 1$)* |
| Activation / output bias | ReLU; output bias initialized $\mathcal{N}(0, 0.01^2)$ (std 0.01) |
| Optimizer | Adam (`optax`) |
| Base (pretraining) learning rate | $10^{-3}$ *(P2P-Dynamics: $5 \times 10^{-4}$)* |
| Model selection | best of 10 independent runs, chosen by validation loss |
| Meta inner / outer learning rate | $\alpha = 0.01$ / $\beta = 0.001$; single inner gradient step |
| Tasks per meta-batch | 16 *(P2P-Cost-Small meta-training: 8)* |
| Evaluation interval | every 100 iterations |
| Input normalization | Zero-mean / unit-variance, statistics from the training set *(P2P-Dynamics: min–max, applied inside a NormalizedSetONet wrapper)* |
| Train / test split | 80%/20% over tasks; seed 42 |
| Loss | MSE / behavioral cloning (Eq. 9); reported as relative $L^2$ error (Eq. 19) |
| Hardware | Single NVIDIA GPU (RTX 5070) |

Table 4: Per-environment architecture and branch/trunk input–output dimensions ($d_x/d_u$ are state/control dimensions). Depth is the number of hidden layers, equal in $\phi$, $\rho$, and the trunk. The trunk input is always $(\mathbf{x}, t)$ and the output is always the control / action.

| Environment | $p$ | Depth | Attn. tokens | Branch location | Branch value | Output dim |
|-------------|-----|-------|--------------|-----------------|--------------|------------|
| P2P-Cost / -Small | 128 | 2 | 4 | $(\mathbf{x}, \mathbf{u})$, 4+2 | cost $\ell$, 1 | 2 |
| P2P-Dynamics | 32 | 4 | 1 | $(\mathbf{x}, \mathbf{u})$, 4+2 | next state, 4 | 2 |
| Quadrotor | 64 | 3 | 4 | $(\mathbf{x}, \mathbf{u})$, 6+2 | next state, 6 | 2 |
| Obstacle | 128 | 4 | 4 | $(x, y)$, 2 | radius $r$, 1 | 2 |
| HalfCheetah-v3 | – | – | – | $(\mathbf{x}, \mathbf{u})$, 17+6 | next state, 17 | 6 |

## A.1 Pretraining (behavioral cloning)

Table 5: Per-environment pretraining settings (shared values in Table 3). $M$ is the number of tasks per batch, $K$ the branch context size (a single fixed size for P2P-Cost/-Small and Quadrotor; a set sampled per iteration for P2P-Dynamics and Obstacle), and $N$ the number of expert trajectories in the loss. For Obstacle the branch context is always the full obstacle set, so "–" denotes no fixed $N$ and its $K$ instead counts the expert trajectories in the loss.

| Environment | Iterations | $M$ | $K$ | $N$ | Learning rate |
|-------------|------------|-----|-----|-----|---------------|
| P2P-Cost | 7500 | 32 | 128 | 32 | $10^{-3}$ |
| P2P-Cost-Small | 7500 | 32 | 128 | 10 | $10^{-3}$ |
| P2P-Dynamics | 15000 | 32 | $\{16, 32, 64, 128\}$ | 16 | $5 \times 10^{-4}$ |
| Quadrotor | 2000 | 16 | 64 | 5 | $10^{-3}$ |
| Obstacle | 20000 | 16 | $\{32, 64, 128\}$ | – | $10^{-3}$ |

## A.2 Meta-training and the MAML baseline

Two MAML-style variants are defined over the SetONet operator: **SetONet-Meta** updates the branch only in the inner loop (trunk frozen), and **SetONet-Meta-Full** updates all parameters in the inner loop (second-order). The **MAML (MLP)** baseline applies the same MAML bi-level objective to a monolithic MLP policy that maps state (and time) directly to the action, with no branch–trunk decomposition and no context-set encoder; task identity is acquired only through inner-loop gradient steps on the support set. All three share the inner/outer learning rates, a single inner step, the Adam outer optimizer, the meta-batch size, and the per-environment iteration budget (Table 3, Table 6); they differ only in the policy class and

which parameters the inner loop updates. The MLP depth is set per environment so that the baseline's parameter count is comparable to the operator.

Table 6: Meta-training and MAML-baseline settings. Both use the same per-environment iteration budget (shown once). For meta-training, the inner-loop support and query sets use the same sizes (Obstacle uses 10 evaluation batches for the query). Shared constants ($\alpha$, $\beta$, width, tasks/batch) are in Table 3.

| | | Meta-training | MAML (MLP) baseline | | |
|---|---|---|---|---|---|
| Environment | Iterations | support (=query) | depth | support | query |
| P2P-Cost | 5000 | $K=64, N=4$ | 16 | $K=64$ | $K=8$ |
| P2P-Cost-Small | 500 | $K=64, N=4$ | 16 | $K=64$ | $K=8$ |
| P2P-Dynamics | 1500 | $K=32, N=8$ | 17 | $N=8$ | $N=8$ |
| Quadrotor | 1000 | $K=64, N=5$ | 17 | $K=64$ | $K=32$ |
| Obstacle | 2000 | $K=10$ | 22 | $K=8$ | $K=8$ |

### A.3 Data generation

Table 7: Data-generation settings. "Traj./task" is the number of expert trajectories per task.

| Environment | Expert | #Tasks | Traj./task | Horizon | $\Delta t$ | Task ranges |
|---|---|---|---|---|---|---|
| P2P-Cost | LQR | 500 goals | 100 | 50 | 0.1 | goal/state $\in [-10, 10]^2$, vel $\in [-5, 5]$ |
| P2P-Cost-Small | LQR | 50 goals | 10 | 50 | 0.1 | as P2P-Cost |
| P2P-Dynamics | iLQR | 100 cfgs | 100 | 30 | 0.1 | friction $\in [0.3, 0.9]$, $v^{\max} \in [10, 15]$, $a^{\max} \in [3, 5]$ |
| Quadrotor | iLQR | 100 cfgs | 20 | 100 | 0.02 | mass $\in [0.1, 0.5]$, arm $\in [0.05, 0.15]$, inertia-scale $\in [0.8, 1.2]$ |
| Obstacle | IPOPT (CasADi) | 500 | $30 \times 2$ | 50 | – | $n_{\text{obs}}$ up to 12, $q_f = q_p = 1000$ |
| HalfCheetah-v3 | SAC | 53 cfgs | 100 | 100 | – | varies mass, limb, joints, friction (external) |

**Cost weights.** P2P-Cost: $Q = 1.0$, $R = 0.1$, $Q_f = 10.0$. P2P-Dynamics: position 1.0, velocity 0.5, control 0.01, terminal position 20.0, terminal velocity 10.0. Quadrotor: position 10.0, velocity 1.0, angle 50.0, angular velocity 5.0, control 0.1, terminal position 100.0, terminal velocity 50.0, terminal angle 100.0. Solver caps: P2P-Dynamics iLQR `max_iter`= 100; Quadrotor iLQR `max_iter`= 150, `max_torque`= 0.1.

### A.4 Task-specific adaptation

All strategies use the pretrained SetONet as a warm start and differ only in which parameters are unfrozen: **SetONet-FT** (full network), **Full-Branch** (entire branch; trunk frozen), **Last-Branch** (final branch layer), **Last-Trunk** (final trunk layer), and **Last-Both** (final layer of each). Adaptation is reported at 0, 1, and 25 gradient steps (Table 2). Adaptation with expert demonstrations minimizes the imitation loss (Eq. 10) on a small support set. Cost-based adaptation (Eq. 11) uses no expert data: the operator is fine-tuned by differentiating the control objective through a differentiable rollout. The Obstacle surrogate cost uses $w_{\text{coll}} = 10$, $w_{\text{ctrl}} = \Delta t$, collision sharpness $\alpha = 15$, and safety margin $m = 0.2$.

## B Context-Set Construction

This section describes how the context set is generated for each environment, what its values physically represent, and what a practitioner would need to construct it on a real system. It is intended to make concrete the practical cost of applying the approach, which is a leading motivation of the paper.

### B.1 What the context set is

The branch network consumes a context set $C = \{(\text{location}_j, \text{value}_j)\}_{j=1}^m$: an unordered, variable-sized collection of pointwise samples that identifies the task without ever exposing the task parameters $(\phi, \psi)$ to the model. The branch maps this set to the task-dependent coefficients $\{c_k\}$; the trunk maps a query $\mathbf{y} = (\mathbf{x}, t)$ to basis functions $\{b_k(\mathbf{y})\}$; the predicted control is $\hat{\pi}(\mathbf{y}) = \sum_k c_k b_k(\mathbf{y})$. Because the branch is

a permutation-invariant set encoder, the context can have any size and ordering at train or test time (the task-resolution invariance of Section 5.2): a practitioner does not need a fixed sensor grid, only the ability to produce *some* collection of (location, value) samples that characterize the task. The pairs fall into three qualitatively different regimes (Table 8).

Table 8: Context-set construction by regime. None of the three requires solving the optimal control problem at deployment.

| Context type | Location | Value | What is needed to produce it |
|---|---|---|---|
| Cost evaluations (P2P-Cost) | $(\mathbf{x}, \mathbf{u})$ | cost $\ell$ | evaluate the cost at chosen points |
| Dynamics transitions (P2P-Dyn., Quadrotor, HalfCheetah) | $(\mathbf{x}, \mathbf{u})$ | next state $\mathbf{x}'$ | observe / log system transitions |
| Task geometry (Obstacle) | obstacle $(x, y)$ | radius $r$ | observe the scene (perception) |

## B.2 Per-environment details

**P2P-Cost / P2P-Cost-Small (cost evaluations).** A context point is location $= (\mathbf{x}, \mathbf{u}) \in \mathbb{R}^6$, value $= \ell(\mathbf{x}, \mathbf{u}; \mathbf{x}_g) \in \mathbb{R}$. During training the points are sampled analytically on the fly: states are drawn uniformly over the workspace and controls uniformly over their range, and the value is the known quadratic stage cost $\ell = (\mathbf{x} - \mathbf{x}_g)^\top Q (\mathbf{x} - \mathbf{x}_g) + \mathbf{u}^\top R \mathbf{u}$ evaluated at those points. An equivalent context can be built from logged data by sampling $(\mathbf{x}, \mathbf{u}, \ell)$ tuples from stored trajectories; either route yields the identical format. The number of context points $K$ is sampled per iteration from $\{32, 64, 128\}$. *Practitioner cost: low — only a queryable cost model is needed; no solver, rollouts, or expert.*

**P2P-Dynamics / Quadrotor (dynamics transitions).** A context point is location $= (\mathbf{x}, \mathbf{u})$, value $= \mathbf{x}'$ (the next state). Context is built by enumerating one-step transitions $(\mathbf{x}_t, \mathbf{u}_t, \mathbf{x}_{t+1})$ from the task's trajectories and sampling $K$ at random ($K \in \{32, 64, 128\}$ for P2P-Dynamics, $K = 64$ for Quadrotor). *Practitioner cost: moderate* — one must observe transitions of the system under the new dynamics (a short interaction log or simulator rollout). Crucially these are transitions, not expert demonstrations: any control sequence that excites the dynamics suffices.

**Obstacle (task geometry).** A context point is location $= (x, y)$ (an obstacle center), value $= r$ (its radius). The context is the obstacle field itself, read directly from the task definition: one context element per obstacle, so the set size equals the number of obstacles ($n_{\text{obs}} \in \{2, 4, 6\}$ in training, generalizing to 3 and 5 at test time, per Section 5.2). No trajectory sampling is involved for the context. *Practitioner cost: low* — a direct observation of scene geometry, exactly what an onboard perception system would provide. This illustrates that the context need not be a smooth cost: it can be a finite-dimensional proxy for the constraint structure.

**HalfCheetah-v3 (logged transitions).** A context point is location $= (\mathbf{x}, \mathbf{u}) \in \mathbb{R}^{23}$, value $= \mathbf{x}' \in \mathbb{R}^{17}$. Transitions are taken from logged rollouts of an SAC expert for each task configuration (varying mass, limb length, joint range, friction), using the same transition-context pipeline as the other dynamics-varying environments. *Practitioner cost: moderate* — a log of state–action transitions for the new agent configuration.

## B.3 Context at training versus deployment

During training, $K$ context points are sampled per iteration, with $K$ itself drawn from a set so the branch never overfits to a fixed cardinality. At deployment on a new, held-out task, the operator receives a single context set — sampled cost evaluations, observed transitions, or the observed obstacle field — and predicts the policy at arbitrary query states in one forward pass, without re-solving the optimal control problem (Section 4.1). If accuracy is insufficient, the same context (optionally with a few demonstrations or the known cost) drives the lightweight adaptation strategies of Section 4.2.

### B.4 Practical considerations

The three regimes map onto increasing real-world effort: a cost-evaluation context needs only a queryable cost model (often the very thing the practitioner is designing); a transition context needs the ability to observe the system's transitions, which can come from a simulator, a system-identification dataset, or routine operational logs, since the transitions are exploratory rather than expert; and a geometry context needs a perception of the task configuration, typically already available from sensors. In no case does constructing the context require solving the parametric optimal control problem, nor (except where explicitly studied in Section 4.2) expert demonstrations for the new task. The open practical challenge is the *informativeness* of the context: it must sufficiently characterize the task (e.g., transitions must excite the relevant dynamics modes). The resolution-invariance study (Section 5.2, Figure 5) gives empirical guidance, roughly 16–32 samples suffice in the control environments before error plateaus, and the dynamics-varying tasks need only a handful of transitions to identify the parameters.

