# OpenReview forum: "Neural Operators for Multi-Task Control and Adaptation"
_TMLR — Under review for TMLR_

### Review · Reviewer_YGmC · 2026-06-08

**Summary Of Contributions:**

This paper studies the application of neural operators, specifically the SetONet/DeepONet variety, on multi-task control. The paper mostly focuses on the offline setting using imitation learning objectives, with some exploration using a Monte Carlo style objective.
_________
Aside:

Before digging into the review, it is worth mentioning briefly how DeepONet architectures work as it is not a commonly used approach in the area. The DeepONet architecture, in the family of neural opeartors, aims to learn a mapping between two infinite-dimensional function spaces rather than mapping from, for instance, just the current (or history of) observation(s) of the control task to the next action taken. In the DeepONet architecture, this is accomplished through separating the input into task-specific input and input signifying the where to query the output function (Figure 2 in the paper shows this). These inputs are fed to parallel neural networks (Branch and Trunk respecitvely) and the final output is an inner product of the two network's outputs. In the literature, being able to use any amount of data in the input function and query at specific locations in the output function is known as "resolution invariance", i.e., meaning the function should be valid using an input size of 5 points or 10 points evaluated at irregular intervals and evaluated at any point in the output function.
_________

What sets this style of architecture apart is its use of extra information that may be available before the control of the system begins. In a real setting, the information may not have a standard amount across all tasks and thus a resolution-invariant architecture could potentially take advantage of these real tasks. The main algorithmic contribution of this paper is in how they enable the architecture to learn for fast adaptation. The paper explores the architecture's innate generalization, how it performs in a pre-train fine-tune regime, and a meta learning with few-shot gradient updates. They also take advantage of the structure of the architecture to limit which parameters are updated smartly and compare across full-network fine-tuning, branch-only fine-tuning, and last-layer fine-tuning (including last in both networks, just the branch, and just the trunk). Finally, they also suggest a meta-learning regime inspired by MAML for deeponet style architectures.


The paper evaluates the architecture's performance on 1) task resolution invariance (i.e., sample invariant for defining the task), 2) multi-task adaptation, and 3) in a meta-learning regime using MAML as influence. The paper evaluates the architecture and several variants in several well known optimal-control systems and half-cheetah from iMuJoCo (Patacchiola et al., 2023). Most comparisons are made through evaluating the predicted trajectories with expert trajectories (i.e., imitation learning cost), with the exception of the experiments using the monte-carlo style rollouts.


Claims:
1. Neural operators are effective for multi-task control, accurately approximating the "solution operator" over a 1) distribution of tasks, and 2) the resolution of data the controller has access to at test time.
2. Taking advantage of the structure of a DeepONet style architecture can enable more efficeint adaptation strategies, specificaly partial fine-tuning is comparable to full-network fine-tuning.
3. "Cost-based fine-tuning" (i.e., fine-tuning through Monte Carlo rollouts) allows adaptation without any explicit expert demonstrations
4. The two approaches introduced for metalearning with DeepONet style architectures outperform MAML on the environments tested in the paper.

**Additional Comments:**

Overall, I think this paper likely can eventually be accepted to TMLR with some work to correct for the errors I mentioned above.

Some other comments:
- There might be missing supplementary somewhere, so I'm more than willing to look into it if provided.
- Overall, the paper is well written and no editing mistakes jumped out to me.
- I think the trajectory plots are often confusing to read.  Make sure you explicitly state what the symbols in the plots are (i.e., stars are goals and dots are starting positions).

**Audience:**

Yes

**Audience Explanation:**

Yes, this would generally be interesting to an audience within the TMLR audience.

Somewhat related recent multi-task articles in TMLR:
- https://openreview.net/forum?id=zIDGm96xwg
- https://openreview.net/pdf?id=9L4Z23EfE9
- https://openreview.net/forum?id=HJbcwRbMQQ

Related recent neural operator articles in TMLR:
- https://openreview.net/forum?id=0S1LWZHQYn
- https://openreview.net/forum?id=9vEVeX9oIv

**Claims And Evidence:**

No

**Claims Explanation:**

### Algorithmic claims and contributions

Most of the claims/contributions related to algorithmic development are all well supported. Specifically, the fine-tuning parameter restrictions are discussed in section 4.2, and the MAML style meta-learning approaches were introduced in section 4.3. While the meta-learning approaches for the deeponet are heavily influenced by MAML with slightly different objectives and restrictions to learned weights between loops (from my reading), this is enough for an interesting set of approaches to compare for TMLR. As far as I'm aware, there are no papers that discuss these particular points and do a thorough comparison across these architectures. I would suggest the authors consider [1] as related work, as they discuss multi-task deeponet style architectures for usual PDE solutions (not control).

While most of the claims and contributions are well positioned from the literature, The rollout idea discussed in section 4.2 "Adaptation with cost feedback" requires better positioning wrt the literature. This is effectively a Monte Carlo style update, but no works are referenced that suggest as such. Related works would be in the line of research related to Offline-to-Online RL (i.e., offline is the imitation learning, then online is the RL component through Monte Carlo updates). Some related work is provided in section 6, but perhaps [2] is a reasonable starting point to search for literature in a more modern context.


### Empirical claims

**Note**: The paper states that architectural and hyperparameter details are in supplementary. I could not find this on the open review, so I assume it was not provided and thus not reviewed. The following review is based on the assumption that this is accurate upon future revisions.


- **Claim 1:** This is partially supported by the evidence of errors wrt expert demonstrations Table 2 (the 0 steps section), by visual inspection of trajectories in Figure 4, and task resolution invariance in figure 5.
- **Claim 2:** This is partially supported in Table 2 (the later sections) comparing pretrained to "SetONet-FT", "Last-Branch", and "Last-Both".
- **Claim 3:** This is partially supported through Figure 7
- **Claim 4:** this is partially supported through Table 2 comparing "MAML", "SetONet-Meta", and "SetONet-Meta-Full" across all sections, figure 6, figure 8, figure 9, and figure 10.


While each claim is supported by some empirical evidence, there are some errors in the analysis that are worth investigating.
- E-1. Table 2: Many of the claims are supported by evidence in table 2. This table has no confidence intervals, or an indication that the comparison was run for multiple training seeds.
- E-2. MAML as a baseline: While I agree MAML is relevant here to compare for claim 4, I'm a bit confused on how it is being applied. MAML stands for Model-Agnostic Meta-Learning, and so it is model agnostic (that means SetONet-Meta-Full is indeed MAML). My belief is that the model labeled MAML is using a different architecture. What is that architecture, and why was it chosen as a reasonable baseline here? Much more justification is needed here.
- E-3. For the environments you use, it is unclear to me how the context set is generated in each. This is an important topic as it relates to practitioners who might be interested in such an approach, as it would be an indication on how difficult it will be to use these ideas in practice (which is a leading motivation of the paper). This seems like a big challenge for applying these methods in the real world (but does not have to be solved by this paper, just needs to be clear).
- E-4. From a control perspective, I understand what you are adding by comparing trajectories visually, but in the context of this paper I'm not sure it adds much. What would be a more interesting comparison would be to show the relative costs of the approaches proposed in this paper with the expert trajectory costs. Much like offline-to-online RL. I think the error wrt the expert trajectory is interesting in some contexts, but less so here. Especially when looking at HalfCheetah-v3. In the end, for imitation learning and reinforcement learning we care about the behavior of the system not the absolute error wrt to specific trajectories. Specifically, Figure 4 and 9 don't tell me much as a reader, but I very much like Figure 7 as it relates the trajectories to the environment so I can understand the behavior a bit better.
- E-5. I think using a standard feed-forward neural network as a baseline in some of the tasks would be beneficial to understand what the SetONet is bringing to the table in-terms of performance benefits. Currently, as you are using mostly new tasks it is not clear how this would relate to other architectures used in multi-task learning. There would be several compromises you will have to make to make such a comparison meaningful.


[1] Kumar, Varun, et al. ["Synergistic learning with multi-task DeepONet for efficient PDE problem solving."][https://www.sciencedirect.com/science/article/pii/S0893608024010426?casa_token=87Xn-XR5uDkAAAAA:5xGNSFIRiHPOSaKvBPkvFsDcnPbbh2o5TziNs77jDYo3Py7dNpzNzAMS720AdLQnmyzbODTL4-Nl] Neural Networks 184 (2025): 107113.

[2] Ball, Philip J., et al. ["Efficient online reinforcement learning with offline data."][https://proceedings.mlr.press/v202/ball23a.html] International Conference on Machine Learning. PMLR, 2023.

**Requested Changes:**

1. I would like to see the "Adaptation with cost feedback" better positioned in the literature early on. This idea is orthogonal to the rest of the paper, but can still be included as an interesting test case using the neural operators.
2. The errors (E-[1-3]) need to be addressed.
3. I would like E-4 and E-5 to be addressed (with a neural network baseline and comparisons on the total cost of the behavior), but I do not see these as critical for this paper to be eventually accepted at TMLR.

---

> ### Author Response · Authors · 2026-07-01
> **Responding to points E1-E3**
>
> We thank Reviewer YGmC for the careful and constructive review, and for noting that the work would be of interest to the TMLR audience. The review centered on empirical rigor and on the positioning of cost-based fine-tuning, and we have revised the paper to address each point. In summary, we have (i) updated Table 2 to report mean ± standard deviation over five seeds [E-1]; (ii) specified the MAML baseline architecture and clarified its relationship to SetONet-Meta-Full, with full hyperparameters now in a dedicated appendix [E-2]; (iii) added a per-environment account of how the context set is constructed, added a new "Context" column in Table 1, a paragraph in Section 5.1, and a dedicated appendix [E-3]; and (iv) repositioned cost-based fine-tuning within the offline-to-online literature and clarified its gradient mechanism. We have additionally included the complete architecture and hyperparameter details, both as a new appendix and as documented supplementary code, which should resolve the difficulty the reviewer noted in locating these materials during review.
>
> # E-1
>
> We agree, and have updated Table 2 to report relative L² error as mean ± standard deviation over five seeds. The reported numbers were already averaged over five seeds; we had shown only the means for compactness, and the reviewer is right that including the spread gives a more complete picture. We have revised the Section 5.3 text accordingly, and everything needed to regenerate the table is in the supplementary code.
>
> # E-2
>
> We thank the reviewer for catching this ambiguity, and the reviewer is correct: applying the MAML algorithm to our operator is exactly SetONet-Meta-Full. The row labeled "MAML" is the standard MAML baseline applied to a different policy class, implemented as a monolithic MLP.
> Concretely, the MAML (MLP) baseline is a feed-forward MLP policy mapping state (and time) directly to the control action, with no branch–trunk decomposition and, crucially, no context-set encoder. Both methods use the identical MAML bi-level objective and optimization: a single inner gradient step, inner learning rate 0.01, outer learning rate 0.001, Adam outer optimizer, 16 tasks per meta-batch, and the same per-environment iteration budget as SetONet-Meta-Full. The MLP has hidden width 128 with depth set per environment (16 layers for P2P-Cost and P2P-Cost-Small, 17 for P2P-Dynamics and Quadrotor, 22 for Obstacle), this ensures that its parameter count is comparable to the neural operator method; the performance gap is therefore not a capacity artifact.
> The essential difference is how each method receives task information:
> The MAML (MLP) baseline has no mechanism to ingest task observations directly; it acquires task identity only through inner-loop gradient steps on the support set. With zero gradient steps it can only output the shared meta-initialization, which is why its zero-shot error in Table 2 is high.
> The SetONet operator encodes the task through the branch network's context set, producing a task-specific policy in a single forward pass (hence strong zero-shot performance); the meta-trained variants additionally refine through inner-loop adaptation.
> We chose this baseline because a monolithic policy network adapted by MAML is the canonical gradient-based meta-learning approach for control and the natural comparison for our method. Holding the meta-algorithm, learning rates, inner-step count, meta-batch size, and compute budget fixed isolates the contribution of the operator architecture itself.
>
>
> # E-3
>
> We agree that our original explanation of how the context is sampled in each environment was not thorough enough and we thank the reviewer for pointing this out. The format of the context per environment is now specified in Sections 4.1 and 5.1, and Table 1 now includes a "Context" column giving the (location → value) pairs for each environment. We have additionally added (i) a "Constructing the context set" paragraph in Section 5.1 describing how the context is sampled in each environment and what a practitioner must produce, and (ii) a dedicated appendix (Appendix B, "Context-Set Construction") with the full per-environment generation procedure, we also include necessary implementation details for context generation in the supplementary code. In brief, the context is built from a queryable cost (P2P-Cost), from observed or logged transitions that need only excite the dynamics rather than be expert-optimal (P2P-Dynamics, Quadrotor, HalfCheetah), or from a direct observation of the scene geometry (Obstacle); in no case does constructing the context require solving the control problem at deployment. We also note that the resolution-invariance study (Section 5.2, Figure 5) quantifies how much context is needed; roughly 16–32 samples before error plateaus, and only a handful of transitions to identify the dynamics-varying tasks.

---

> ### Author Response · Authors · 2026-07-01
> **Responding to E4-E5**
>
> # E-4
>
> We agree that achieved behavior is ultimately what matters, and we gave this careful consideration. Our metric, relative L² error against the expert control, is the standard measure for behavioral-cloning and operator-approximation tasks: it is scale-consistent across environments of differing magnitudes, and it directly measures how well the operator approximates the target solution operator, which is the central claim of the paper. We also note that our evaluation already goes beyond pointwise deviation on fixed expert states: the model-rollout results (Section 5.2, Figure 6) execute the learned policy in closed loop from the initial state without expert feedback, and thus test achieved behavior rather than open-loop control error. Figure 7 similarly relates trajectories directly to the task objective and environment (distance-to-goal for P2P-Cost, collision time for Obstacle), which we believe captures the behavioral quality the reviewer is interested in for the settings where a task-level metric is most meaningful. We explored reporting closed-loop cost normalized to the expert as an additional metric, but found that such comparisons introduce confounds specific to closed-loop rollouts, particularly stability margins and sensitivity to a small number of divergent trajectories, that make cross-method attribution difficult and can obscure rather than clarify approximation quality. We therefore retain relative L² as our primary metric, complemented by the rollout and task-level evaluations already in the paper.
>
> # E-5
>
> We have added some baseline results and discussion. We put this in our response to reviewer 1ydV under 1ydV-3.

---

> > ### Comment · Reviewer_YGmC · 2026-07-08
> >
> > Thank you for your answers, edits, and revisions. I think you have cleared up most of the confusions I had in the original document. I particularly appreciate the added context in the appendix for the context set, hyperparameters, and MAML baseline.
> >
> > Just one follow up: You mentioned that you modified the positioning of the cost-based feedback mechanism. Can you point to where this was modified? I can't find it in the document and would like to see this (the original section 4.2 seems unchanged wrt this).

---

> > > ### Author Response · Authors · 2026-07-08
> > > **Responding to "modified the positioning of the cost-based feedback mechanism"**
> > >
> > > Thank you for the response. Your right that there are no changes in section 4.2. The repositioning is in the Related Work, Section 2.1 (Data-Driven Optimal Control): we added a paragraph beginning "While offline imitation amortizes the cost of online optimization..." that frames cost-based fine-tuning within the offline-to-online paradigm, relating it to differentiable predictive control (Drgoňa et al., 2024) and self-supervised operator learning for control (Xu et al., 2025). Our aim there was to better position the mechanism within the existing literature and to make explicit that the update is an analytic policy gradient obtained by backpropagating the control objective through the differentiable rollout, rather than a Monte-Carlo/score-function estimate, which we understood to be the mechanism concern in the original review. Apologies for the confusion in pointing to Section 4.2; the framing paragraph lives in 2.1, while 4.2 defines the mechanism itself.

---

> > > > ### Comment · Reviewer_YGmC · 2026-07-13
> > > >
> > > > Excellent! I would have preferred this be referred to in section 4.2 as well, just to tie it more easily back to the background. This also reinforces that you are taking the gradient through the dynamics, which I had missed in a previous read :), thus my belief this was monte-carlo sampling.
> > > >
> > > > My only preference is you link these two sections together in 4.2, but otherwise I don't think there are any more concerns with the added clarifications for my and the other reviewers concerns.

---

> > > > > ### Author Response · Authors · 2026-07-15
> > > > > **Section 4.2 Edit**
> > > > >
> > > > > Thank you! We agree that linking the two sections directly in Section 4.2 improves the paper. We have uploaded a revised version in which Section 4.2 now reads, immediately after the gradient computation: "This yields an analytic policy gradient: the control objective is differentiated through the dynamics themselves. In this sense, cost-based fine-tuning is not a separate technique but the same adaptation framework with the imitation loss (10) replaced by the control objective (11); this situates it within the offline-to-online line of data-driven optimal control discussed in Section 2.1." This makes explicit both the connection back to Section 2.1 and that the gradient is taken through the dynamics rather than estimated by sampling. Thank you for the constructive review throughout the discussion.

---

### Review · Reviewer_PGZy · 2026-06-10

**Summary Of Contributions:**

This paper proposes applying Neural Operators (specifically SetONet) to multi-task optimal control. The authors formulate multi-task control as learning a continuous mapping from task descriptions to optimal policies. The method uses a branch-trunk structure: the branch encodes variable-sized, unordered task observations, while the trunk maps query states to basis functions. The model is initially trained via behavioral cloning on expert demonstrations. To handle out-of-distribution tasks, the authors propose structured fine-tuning strategies (e.g., branch-only, cost-based) alongside two meta-learning variants for few-shot adaptation. The approach is evaluated on four parametric control environments and HalfCheetah-v3.

**Audience:**

Yes

**Audience Explanation:**

This work will be of interest to researchers in reinforcement learning, optimal control, and scientific machine learning. Applying neural operators to control is an underexplored area compared to PDEs. Showing that a branch-trunk operator can bypass the bottlenecks of standard finite-dimensional task embeddings and consistently outperform MAML in continuous control is a solid contribution for the TMLR audience.

**Broader Impact Concerns:**

This is a methodological paper on optimal control with no obvious societal risks; a dedicated broader impact statement is not necessary.

**Claims And Evidence:**

Yes

**Claims Explanation:**

The empirical evidence largely supports the authors' claims of efficiency, task-resolution invariance, and superiority over MAML. The experiments span linear, non-linear, and high-dimensional RL dynamics effectively. However, the performance claims need slight qualification. As seen in Figure 5, performance degrades on the Obstacle Avoidance task when scaling from 2 to 6 obstacles, suggesting the operator struggles with complex constraint geometries. Furthermore, the pretraining methodology heavily depends on robust expert solvers (LQR, iLQR, SAC). The paper should better address how bounded expert data (like the suboptimal SAC rollouts in HalfCheetah) impacts the quality of the learned basis functions.

**Requested Changes:**

My list of requested changes, together with whether they are critical for acceptance or not, is given below:

1. Please discuss the performance degradation in the Obstacle Avoidance task as constraint complexity increases. Clarify if this is a fundamental limitation of the branch encoder or if scaling network capacity/context points could mitigate it. (critical for acceptance)

2. Address the reliance on expert solvers for pretraining in the limitations section. Specifically, I would like to see a discussion of how noisy or suboptimal expert data impacts the trunk network's ability to learn robust, generalizable basis functions. (critical for acceptance)

3. Clarify the assumptions for cost-based fine-tuning. Please specify if this requires a perfectly known, fully differentiable dynamics model and discuss the implications for learned or stochastic dynamics. (recommended)

4. I would like to see a brief comparison of the initial pretraining computational cost of SetONet versus the baselines to provide a more complete picture of the training trade-offs. (recommended)

---

> ### Author Response · Authors · 2026-07-01
> **Response to requested changes 1-4**
>
> Thank you for your constructive and thorough feedback. We appreciate your recognition of our work's contribution and your valuable suggestions to improve the clarity and depth of our analysis. We address the concerns in the following points.
>
> # 1
>
> We thank the reviewer for the feedback and agree this needs clarification. As the reviewer notes, performance decreases as the number of obstacles (and hence the constraint complexity) increases. However, we believe this reflects the increased problem difficulty, and not any limitation in the branch encoder. In particular, our intention with the obstacle avoidance scenario in Figure 5 was to show generalization. We train on tasks with obstacle counts of {2,4,6} and show that evaluating on tasks with counts of 3 and 5 does not give a sudden jump in the task loss. This supports the interpretation that the drop in performance comes not from any fundamental limitation of the branch encoder, but rather from the inherent increase in problem difficulty.
>
> # 2
>
> We agree that a limitations section would be beneficial here. We would emphasize, however, that limitations due to noisy or suboptimal expert data are not unique to our approach; any method based on imitation learning inherits the same issues. Below we discuss the limits of our approach under suboptimality, and we will add this discussion to a limitations section at the end of the paper.
>
> When the demonstrations used to train the operator are systematically biased, the operator inherits that bias through behavioral cloning: the trunk learns a basis that spans the demonstrated (biased) policy family rather than the optimal one, and additional basis capacity cannot correct a systematically biased target. The quality of the learned basis is therefore upper-bounded by the systematic quality of the experts used for pretraining. To overcome this, objective-based fine-tuning offers a route: because its training signal is the task objective itself rather than the expert's actions, its optimum is the minimizer of that objective, not the demonstrator, so it can refine the operator beyond the expert where a differentiable model is available. Our Obstacle result, in which the cost-adapted operator achieves fewer collisions than the hard-constrained expert (see our response to Reviewer 1ydV, point 2), is a concrete instance. In our experiments, the iMuJoCo setting is the only one with a genuinely suboptimal expert: the SAC policy is learned rather than an optimal solver. There, a behaviorally-cloned operator can at best approach the SAC policy performance (this is what the relative error in Figure 10 reflects), whereas an objective-based signal could in principle surpass it.
>
> # 3
>
> We thank the reviewer; we have clarified these assumptions in the limitations section at the end of the paper.
> Cost-based fine-tuning requires that the total trajectory cost be differentiable with respect to the policy parameters, meaning both the cost function and the transition dynamics must be differentiable. However, this does not require a "perfectly known" analytical model. A learned dynamics model (such as a neural network) is differentiable by construction and can be directly substituted into Equation 11. In this scenario, the adaptation optimizes the modeled objective, meaning systematic model bias could transfer into the adapted policy—a standard trade-off widely studied in model-based reinforcement learning.
> While stochastic optimal control is not the focus of this work, it is compatible with our proposed approach. Although stochastic optimal control problems are generally more challenging to solve, imitation learning remains applicable whenever expert demonstrations or expert solvers are available. For cost-based fine-tuning, the deterministic ODE integrators used in our current implementation (e.g., Euler or RK4) can be replaced with SDE integration methods such as Euler-Maruyama. We refer the reviewer to https://arxiv.org/abs/2601.08594 for a more detailed discussion of SDE integration and adjoints, although we leave a thorough investigation of this direction to future work.
>
> # 4
>
> We have added some baseline results and discussion. We put this in our response to reviewer 1ydV under 1ydV-3.

---

### Review · Reviewer_1ydV · 2026-06-16

**Summary Of Contributions:**

This paper proposes a neural-operator framework for multi-task control and adaptation, where a SetONet model learns to map task descriptions to feedback policies. In Section 1, the authors motivate multi-task control as a mapping from task-defining functions to optimal policies. Section 2 reviews related work on data-driven control, multi-task control, meta-learning, and neural operators. Section 3 formulates the parametric optimal control problem, introduces behavioral cloning, and describes the DeepONet/SetONet architecture. Section 4 presents the main method of the paper, how to adapt, including multi-task behavioral cloning, task-specific fine-tuning with expert demonstrations, cost-based fine-tuning through differentiable rollouts, and two MAML-style meta-training variants. Section 5 evaluates the method on several control benchmarks, including point-to-point control, dynamics-varying systems, quadrotor, obstacle avoidance, and HalfCheetah, focusing on zero-shot prediction, task-resolution invariance, and adaptation to held-out or shifted tasks.

Overall, the paper presents an interesting operator-learning perspective on multi-task control and combines several adaptation mechanisms within a single framework. Its main empirical message is that a SetONet-style policy operator can serve as a reusable controller across a family of related tasks, while branch-trunk fine-tuning and meta-training provide different trade-offs between adaptation cost and flexibility.

$\mathbf{Strength}:$

1. The paper systematically study of adaptation strategies for neural-operator-based control, including full-network fine-tuning, branch/last-layer fine-tuning, cost-based fine-tuning, and MAML-style meta-training to encourage the later fine tuning.

2. The paper studies several adaptation mechanisms, including fine-tuning from expert demonstrations and cost-based adaptation in settings where the task-specific cost function and differentiable dynamics are available.

3. The experiments covers multiple control example, from simple point-to-point systems to quadrotor, obstacle avoidance, and HalfCheetah.

$\mathbf{Questions}:$

1. The paper emphasizes that SetONet does not require direct access to the explicit task parameters $\phi$ and $\psi$, relying instead on finite context samples. However, the cost-based fine-tuning procedure assumes access to the task-specific cost function and differentiable dynamics model.  Do you assume that the downstream task is draw from the same task distribution as the training that using expert? If so, since you already know the analytic form of cost functional and dynamics, why not use physics-informed neural operator loss to train the model? If not, this be out-of-distribution problem, and authors should clarify in what case the cost-based fine-tuning is possible.

2. If the expert is suboptimal, I believe SetONet inherit the expert’s flaws. Can cost-based adaptation improve beyond the expert?

3. Most OCP examples in the experiments appear to share the same running-cost and/or dynamics templates, with only low-dimensional task parameters \phi or \psi varying. In such settings, a natural baseline would be a task-conditioned amortized policy, for example $\pi_\theta(x,t,\phi)$ or $\pi_\theta(x,t,\psi)$. Including such comparisons would help clarify the advantage of the proposed SetONet operator formulation and quantitatively demonstrate when and why the proposed adaptation methods are preferable.


4. The proposed SetONet framework appears closely related to in-context learning: the branch network receives a context set of task observations and produces task-specific coefficients for answering policy queries. Could the authors clarify the distinction between their method and in-context policy learning or context-based meta-learning in the introduction, which I believe could attract more attention.

5. The OOD claim seems somewhat strong. The Quadrotor OOD experiment appears to involve a shifted goal location, but this is only one type of distribution shift. What happens under other shifts, such as unseen numbers of obstacles, more complex obstacle layouts, dynamics parameters outside the training range, or tasks requiring qualitatively different maneuvers such as sharp turns? In harder OOD settings, the learned trunk basis functions may no longer be expressive enough, and full trunk adaptation may be necessary. The paper should clarify the limits of the OOD claim.

6. In Section 4.3, the meta-training objective uses a support/query split within each task, which is standard for MAML-style adaptation. Formally, the procedure itself looks like encouraging the fine-tuning for one fixed task. However, I am not sure why such method would encourage in-distribution task generalization or out-distribution generalization, and I believe the generalization is one of the main goals of this proposed method.

7. In Figure 7b, why does the pretrained SetONet have such a high collision rate on obstacle avoidance? Four-obstacle avoidance should be relatively easy, especially since the context explicitly contains obstacle geometry. Is the pretrained model trained only by behavioral cloning, or does it also use cost or constraint information during pretraining?

**Audience:**

Yes

**Audience Explanation:**

TMLR readers interested in operator learning, imitation learning, meta-learning, and adaptive control would likely find the proposed SetONet-based adaptation framework relevant.

**Broader Impact Concerns:**

No.

**Claims And Evidence:**

Yes

**Claims Explanation:**

Yes, the main claims are generally supported by the empirical evidence, but several claims would benefit from stronger clarification or additional experiments. Please see my comments above for details.

**Requested Changes:**

I would appreciate it if the authors could address the following questions and consider the corresponding revisions.

$\mathbf{Questions}:$

1. The paper emphasizes that SetONet does not require direct access to the explicit task parameters $\phi$ and $\psi$, relying instead on finite context samples. However, the cost-based fine-tuning procedure assumes access to the task-specific cost function and differentiable dynamics model.  Do you assume that the downstream task is draw from the same task distribution as the training that using expert? If so, since you already know the analytic form of cost functional and dynamics, why not use physics-informed neural operator loss to train the model? If not, this be out-of-distribution problem, and authors should clarify in what case the cost-based fine-tuning is possible.

2. If the expert is suboptimal, I believe SetONet inherit the expert’s flaws. Can cost-based adaptation improve beyond the expert?

3. Most OCP examples in the experiments appear to share the same running-cost and/or dynamics templates, with only low-dimensional task parameters \phi or \psi varying. In such settings, a natural baseline would be a task-conditioned amortized policy, for example $\pi_\theta(x,t,\phi)$ or $\pi_\theta(x,t,\psi)$. Including such comparisons would help clarify the advantage of the proposed SetONet operator formulation and quantitatively demonstrate when and why the proposed adaptation methods are preferable.


4. The proposed SetONet framework appears closely related to in-context learning: the branch network receives a context set of task observations and produces task-specific coefficients for answering policy queries. Could the authors clarify the distinction between their method and in-context policy learning or context-based meta-learning in the introduction, which I believe could attract more attention.

5. The OOD claim seems somewhat strong. The Quadrotor OOD experiment appears to involve a shifted goal location, but this is only one type of distribution shift. What happens under other shifts, such as unseen numbers of obstacles, more complex obstacle layouts, dynamics parameters outside the training range, or tasks requiring qualitatively different maneuvers such as sharp turns? In harder OOD settings, the learned trunk basis functions may no longer be expressive enough, and full trunk adaptation may be necessary. The paper should clarify the limits of the OOD claim.

6. In Section 4.3, the meta-training objective uses a support/query split within each task, which is standard for MAML-style adaptation. Formally, the procedure itself looks like encouraging the fine-tuning for one fixed task. However, I am not sure why such method would encourage in-distribution task generalization or out-distribution generalization, and I believe the generalization is one of the main goals of this proposed method.

7. In Figure 7b, why does the pretrained SetONet have such a high collision rate on obstacle avoidance? Four-obstacle avoidance should be relatively easy, especially since the context explicitly contains obstacle geometry. Is the pretrained model trained only by behavioral cloning, or does it also use cost or constraint information during pretraining?

---

> ### Author Response · Authors · 2026-07-01
> **Responding to Q1 - Q2**
>
> We thank Reviewer 1ydV for the thorough and technically engaged review. Their points center on the assumptions and positioning of cost-based fine-tuning, the mechanics of adaptation and meta-training, and the scope of our out-of-distribution claim; we address each in turn below.
>
> # Q1
>
> Thank you for the clarifying question about our assumptions for cost based fine tuning. The method assumes that you have access to the differentiable dynamics and are able to backpropagate through a rollout. This does require access to the accurate gradients of cost and dynamics with respect to the model parameters. The method itself however, does not make an assumption about where the downstream task is coming from. Referring to our obstacle avoidance example (Fig 7.b), you could fine tune on a task with a higher obstacle count or arrangement than what you saw during training.
>
> We also note this does not conflict with the operator's context-based task identification. The pretrained operator identifies a task from context samples without access to the explicit parameters (φ, ψ), and its zero-shot and demonstration-based adaptation paths require neither a cost function nor a dynamics model. Cost-based fine-tuning is an additional, optional refinement available when a differentiable cost and model happen to be at hand; it does not change the requirements of the base method.
>
> Relation to a physics-informed / objective-based loss. We agree this is the right connection to draw, and in fact cost-based fine-tuning already is an objective-based, self-supervised loss in the sense the reviewer suggests: rather than fitting expert labels, it minimizes the control objective directly using the known differentiable dynamics (the same mechanism that underlies differentiable predictive control, Drgoňa et al., 2024, which we now cite). The design question is therefore not whether to use such a loss, but a trade-off: we use behavioral cloning for pretraining because it is stable, inexpensive, and yields a strong warm start across the whole task distribution in a single stage, and we use the objective-based loss for per-task refinement. Pushing the objective/residual loss into pretraining as well is a reasonable alternative and a natural extension, but it does carry practical costs. For example, backpropagation through long rollouts can suffer from ill-conditioning (Metz et al., 2021), especially when underlying dynamics are stiff. Objective/residual losses introduce term-weighting difficulties, and they presume a fully differentiable model is available at training time for every task, which may not hold true especially in robotics and contact-rich environments. Our two-stage design (a cheap label-based warm start, with objective-based refinement only where needed) is a deliberate middle ground. Lastly because our approach decouples the pretraining and cost based fine tuning users are less restricted at inference time.
>
> # Q2
>
> We agree that, like any behavioral cloning approach, the performance of the pre-trained operator is fundamentally bounded by the quality of the demonstrations. However, we emphasize that this limitation stems from the imitation learning objective itself rather than the SetONet architecture.. In practice, though, the severity depends on the quality of the expert, and for most of our environments that quality is high. For the classical control problems (P2P-Cost, P2P-Dynamics, Quadrotor, Obstacle), expert solvers such as LQR, iLQR, and constrained nonlinear programming are fast and accurate, and provide at- or near-optimal references. There is therefore little suboptimality to inherit in these environments. The one genuinely suboptimal expert in our experiments is in the iMuJoCo setting, where the SAC expert is a learned policy rather than an optimal solver. True optimal solutions for the complex iMuJoCo examples are not obtainable, and we observe the SAC solver to be a stable expert across seeded trials.
>
> The cost based fine tuning is one approach to improving upon a suboptimal expert. The pretrained model might inherit the experts bias, but fine tuning on an objective that is independent of the expert  can overcome the original bias. Because its training signal is the task objective itself rather than the expert's actions, its optimum is the minimizer of that objective, not the demonstrator. In our experiments the iMuJoCo setting is the only place we have sub optimal experts, where the SAC expert is a learned, potentially suboptimal policy rather than an optimal solver. There, a behaviorally-cloned operator can at best approach the SAC policy (this is what Figure 10 measures), whereas an objective-based signal could in principle surpass it.

---

> ### Author Response · Authors · 2026-07-01
> **Responding to Q3 - Q4**
>
> # Q3
>
>
> | Method | P2P-Cost | P2P-Dyn. | Quadrotor |
> |---|---|---|---|
> | SetONet (pretrained) | 0.055 ± 0.002 | 0.098 ± 0.001 | 0.064 ± 0.001 |
> | MLP, context (B2) | 0.027 ± 0.001 | 0.129 ± 0.002 | 0.084 ± 0.003 |
> | MLP, task params (B1) | 0.023 ± 0.000 | 0.049 ± 0.001 | 0.044 ± 0.001 |
>
> [Baseline Table] Zero-shot relative L² error (mean ± std over 5 seeds), evaluated at fixed K=64 for comparability with the fixed-vector B2 baseline. B1 is handed the ground-truth task parameters (oracle); B2 infers the task from the same context as the operator's branch, flattened into a fixed-length vector.
>
> We have run the task-conditioned baseline suggested by the reviewer, using an MLP with approximately the same number of parameters as the SetONet model. We ran one model (B1) with direct access to the task parameters, and another (B2) with the same information the SetONet branch encoder receives as context. We note that a flattened-vector MLP requires a fixed context cardinality; to enable a fair comparison we therefore fixed the context size to K=64 for both training and evaluation, the only regime in which the B2 baseline is defined. We omit the Obstacle environment because its context is an inherently variable-size set (one element per obstacle); a fixed-vector baseline cannot represent it at all, which itself illustrates one of our approaches main advantage.
> B1 outperforms both SetONet and B2 on every task, but it has access to task information the operator never receives; it is therefore an oracle that quantifies the value of the information the operator must infer, not a competing method. Among the fair comparison, B2 is competitive only in the simplest environment. On P2P-Cost, an LQR problem with a smooth quadratic cost and a low-dimensional, near-linear task-to-policy map, the MLP fits the mapping directly and slightly outperforms the operator (0.027 vs. 0.055). On the more complex environments the comparison reverses: SetONet outperforms B2 on both P2P-Dynamics (0.098 vs. 0.129) and Quadrotor (0.064 vs. 0.084), where the nonlinear task-to-policy map is not well captured by a flattened-context MLP.
> We reemphasize that this comparison understates the capabilities of the operator. The B2 baseline is only defined for a fixed context size chosen at training time: it cannot handle variable-size or unordered context sets and has no notion of permutation invariance. The comparison was therefore run in the baseline's most favorable regime, a single fixed context size, which withholds a central capability of the operator: training at one context resolution and deploying at another (Section 5.2, Figure 5).
> We also note that naive conditioning for handling task variations can often struggle as the size and complexity of each problem increases. These challenges go beyond control-related problems and are well documented in works such as Perez et al. (2017) and Wang et al. (2024).
>
> # Q4
>
> The reviewer is correct that SetONet performs a form of in-context task adaptation: the branch network conditions on a context set of task observations and produces a task-specific policy in a single forward pass. Rather than distancing our method from in-context learning, we think it is most accurate to position it as a structured, operator-theoretic instance of it, and to make two distinctions. We think these distinctions are important and will update our introduction to help clarify.
> First, relative to sequence-model in-context learning (for example the Decision-Pretrained Transformer (Lee et al., 2023)), where a transformer predicts an action from a query state and an in-context dataset of interactions; our conditioning is permutation and resolution-invariant by construction: the branch is a set encoder, so predictions are invariant to the ordering of context points and well-defined for variable context-set sizes (Section 5.2), whereas transformer context encoders are order-dependent and typically trained at a fixed context length.
> Second, relative to context-based meta-RL such as PEARL (Rakelly et al., 2019), which infers a fixed-dimensional latent task variable and conditions a policy on it, SetONet maps the context to coefficients over a learned basis (the trunk) without compressing the task into a fixed-length vector. This avoids the representational bottleneck we discuss in Section 2.2, and inherits the operator universal-approximation guarantees for maps between function spaces (Chen & Chen, 1995).

---

> ### Author Response · Authors · 2026-07-01
> **Responding to Q5 - Q6**
>
> # Q5
>
> We agree our OOD claims require clarification and we thank the reviewer for pointing this out. In this work, we formally define an OOD task as any task whose parameters lie outside the convex hull of the task parameters seen during the pretraining phase. This gives us a precise definition which we use to evaluate whether the model can generalize beyond its exact training boundaries.
> Neural operators such as SetONet benefit from the universal approximation theorem for operators \citep{chen1995universal}, where the learned basis functions span the task-specific Hilbert space, enabling both interpolation and extrapolation in the corresponding space, hence the stable OOD performance observed in our experiments. However, tasks that require policies completely beyond the span of the learned basis expose the fundamental limitations of a pretrained model, as the fixed trunk network bases are insufficient to approximate the desired policy function, hence demand the need for test-time tuning of the learned basis functions. Our quadrotor example is set up for this exact scenario, and as we observe, updating the trunk network is required for accurately solving the new task.
> In our experiments, we limit our evaluation to one class of tasks for testing to ensure the environments are well controlled and results interpretable. Multi-operator generalization in the policy space poses an interesting question, though it is beyond the scope of this paper.
>
> # Q6
>
> Thank you for the question regarding the formulation of the meta objective and whether it is “encouraging the fine-tuning for one fixed task”. We would like to clarify how generalization enters the meta-learning objective. The inner-loop adaptation is a subroutine inside the objective, not the objective itself. The quantity we actually minimize is a bi-level expectation over the task distribution. The inner step produces a task-specific set of parameters \theta’ which are never retained. They are only used to score the shared initialization \theta by how well it adapts. The outer gradient updates \theta based on the scores across a batch of task specific parameters (\theta’) from the inner optimization. The objective is thus not "fit task \tau" but "find an initialization from which a single gradient step generalizes across the task distribution." Two features make this a generalization objective rather than per-task fitting. First, the support/query split: the inner loop adapts on the support set, but the loss that drives the outer update is evaluated on a held-out query set from the same task. This rewards initializations whose post-adaptation policy generalizes to unseen queries, and explicitly penalizes memorizing the support points. Second, the outer expectation is taken over a batch of tasks, i.e. \theta is optimized to adapt well across the whole distribution of tasks, including tasks not seen during meta-training. We grant that any single episode, viewed in isolation, looks like per-task adaptation; generalization is a property of the outer objective averaged over many tasks, not of any individual inner loop.

---

> ### Author Response · Authors · 2026-07-01
> **Responding to Q7**
>
> # Q7
>
> We thank the reviewer; this lets us clarify both the pretraining setup and the source of the collision behavior. We have added clarifying points to the main text, described below.
>
> Yes, pretraining is behavioral cloning only. To answer the direct question: the pretrained SetONet is trained purely by behavioral cloning of the expert's controls (the IPOPT solutions). It does not use the cost or the collision constraint during pretraining. Constraint information enters only at the cost-based fine-tuning stage (the adapted variants in Fig. 7b), through the soft collision penalty. We have stated this explicitly in Section 5.4
>
> *Why BC-only yields high collision time, despite obstacle geometry in the context*. The reviewer is right that obstacle avoidance with a few obstacles is geometrically simple and that the branch network receives the obstacle geometry. The high collision count of the pretrained model is not a sign that the task is hard, but a consequence of three properties of behavioral cloning:
>
> 1. *No feasibility term in the objective*. BC minimizes control-prediction error against the expert; it contains no penalty for collisions. Having the geometry in the context lets the model identify the obstacle layout, but identification is not avoidance; nothing in the imitation loss informs the model that proximity to those obstacles is catastrophic. The model is trained to match the expert's controls on average, not to remain feasible.
> 2. *Compounding error in closed loop* (Ross et al., 2011). Small per-step control errors accumulate over a rollout and push the state off the expert's trajectory distribution; near an obstacle, a small drift becomes a collision. Collision time is a closed-loop, integrated quantity, so it is especially sensitive to such drift: once a trajectory grazes an obstacle, it can remain in collision for many consecutive steps. This is why a model with low open-loop control error can still incur a high closed-loop collision count.
> 3. *A zero-margin (hard-constrained) expert*. The IPOPT expert enforces collision avoidance as a hard constraint and tends to travel along constraint boundaries with little safety margin. Imitating such boundary-hugging trajectories is intrinsically fragile: any imitation error in the obstacle-ward direction crosses into collision. This is precisely why cost-based fine-tuning, which adds an explicit soft proximity margin the expert never optimized, reduces collisions below even the expert (as discussed in our response to point 2).
>
> On the reported magnitude. We note that the Fig. 7b value is an aggregate, not a single-task count: each trajectory's collision count is the sum of in-collision timesteps over the rollout, and the reported 538 is the mean of that quantity over all held-out trajectories, across configurations of varying obstacle count (2, 4, and 6). It is therefore not a "four-obstacle" number; it pools easier and harder layouts. Combined with the three effects above, a mean of a few hundred summed collision-timesteps from a feasibility-unaware BC model is unsurprising.
>
> Finally, it is worth situating this from a deployment perspective. In practice, a learned policy is typically not deployed as the sole guarantor of constraint satisfaction; instead, a safety filter is layered on top of the nominal policy to enforce constraints at execution time \citep{fisac2018general}. Such mechanisms are complementary to the learning approach studied here: they would enforce collision avoidance at deployment regardless of the nominal policy's raw collision rate. We regard this deployment-time constraint enforcement as beyond the scope of the present work, whose focus is the operator-learning and adaptation framework itself, but we note it as the standard practical route to guaranteed constraint satisfaction.